# Guided Score identity Distillation for Data-Free One-Step Text-to-Image Generation

**Mingyuan Zhou**[1,2,*]**, Zhendong Wang**[1]**, Huangjie Zheng**[1]**, and Hai Huang**[3,*]

[1]The University of Texas at Austin, [2]Google DeepMind, and [3]Atlassian

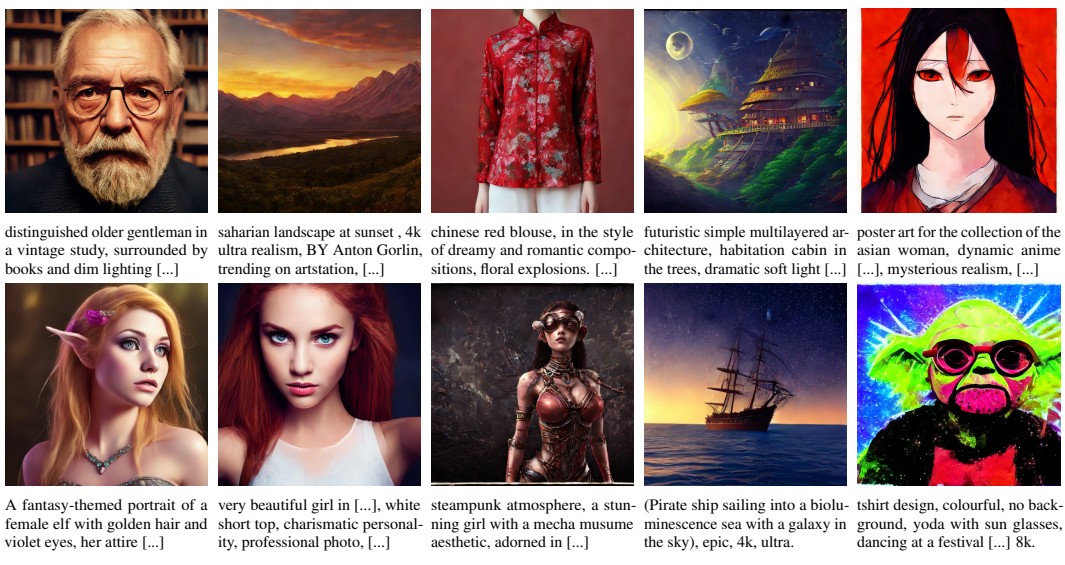

Figure 1: Example generation results of resolution 512x512 from the one-step generator distilled from Stable Diffusion 2.1-base using the proposed method: Score identity Distillation with Long-Short Guidance.

## Abstract

Diffusion-based text-to-image generation models trained on extensive text-image pairs have demonstrated the ability to produce photorealistic images aligned with textual descriptions. However, a significant limitation of these models is their slow sample generation process, which requires iterative refinement through the same network. To overcome this, we introduce a data-free guided distillation method that enables the efficient distillation of pretrained Stable Diffusion models without access to the real training data, often restricted due to legal, privacy, or cost concerns. This method enhances Score identity Distillation (SiD) with Long and Short Classifier-Free Guidance (LSG), an innovative strategy that applies Classifier-Free Guidance (CFG) not only to the evaluation of the pretrained diffusion model but also to the training and evaluation of the fake score network. We optimize a model-based explicit score matching loss using a score-identity-based approximation alongside our proposed guidance strategies for practical computation. By exclusively training with synthetic images generated by its one-step generator, our data-free distillation method rapidly improves FID and CLIP scores, achieving state-of-the-art FID performance while maintaining a competitive CLIP score. Notably, the one-step distillation of Stable Diffusion 1.5 achieves an FID of **8.15** on the COCO-2014 validation set, a record low value under the data-free setting. Our code and checkpoints are available at https://github.com/mingyuanzhou/SiD-LSG.

---

[*]The majority of the work was done while the authors were at Google.

# 1 INTRODUCTION

The pursuit of generating high-resolution, photorealistic images that matches the textual descriptions has driven the machine learning community in developing powerful text-to-image (T2I) generative models. T2I diffusion models (Nichol et al., 2022; Ramesh et al., 2022; Saharia et al., 2022; Rombach et al., 2022; Podell et al., 2024) are currently leading the way in delivering unprecedentedly visual quality, diverse generation, and accurate text-image correspondences. They are renowned for their straightforward implementation and stability during optimization, and they receive substantial acclaim for the robust support from the open-source community (Rombach et al., 2022; Podell et al., 2024).

Despite these advantages, a significant limitation of diffusion models, including those used for T2I tasks, is their slow sampling process, which involves iterative refinement through repeated passes of the generation network. Originally, this required thousands of stochastic sampling steps (Song & Ermon, 2019; Ho et al., 2020; Dhariwal & Nichol, 2021; Song et al., 2021a). Recent advancements in ODE-based deterministic samplers have significantly reduced the required number of sampling steps to just tens or hundreds (Song et al., 2021a; Liu et al., 2022a; Lu et al., 2022b; Karras et al., 2022). To further reduce the number of steps below ten, or even down to one, the focus has shifted toward distilling the iterative-refinement based multi-step T2I generative progress, using a wide variety of acceleration techniques (Zheng et al., 2023b; Meng et al., 2023; Liu et al., 2022b; Luo et al., 2023b; Nguyen & Tran, 2024; Sauer et al., 2023b; Xu et al., 2023; Yin et al., 2023). However, they often result in clearly reduced ability to match the original data distribution, reflected as clearly worsening FIDs. All of them, with the exception of SwiftBrush (Nguyen & Tran, 2024), also require access to real images or the assistance of extra regression or adversarial losses.

It is commonly believed that student models used for distillation sacrifice performance for increased speed. However, recent findings from Score identity Distillation (SiD) (Zhou et al., 2024) present a notable discovery. The SiD-based single-step student model, although trained in a data-free manner, not only simplifies the multi-step generation process required by the teacher diffusion model, EDM by Karras et al. (2022), but also excels in performance. It surpasses the teacher model in terms of Fréchet inception distance (FID) (Heusel et al., 2017) on the CIFAR10-32x32, FFHQ-64x64, and AFHQ-v2 64x64 datasets. It only slightly underperforms in FID in comparison on ImageNet 64x64.

The success of SiD in distilling EDM diffusion models for non-T2I generation in the pixel space has inspired us to adapt it to open-source T2I latent diffusion models, specifically Stable Diffusion (SD) versions 1.5 and 2.1-base, aiming to significantly enhance their generation speed while maintaining performance. However, adapting SiD poses several notable challenges: first, SiD-EDM does not incorporate classifier-free guidance (CFG) (Ho & Salimans, 2022), which is integral to SD; second, SiD has primarily been applied to distill pre-trained EDM models, which utilize noise scheduling and preconditioning methods markedly different from the DDPM noise scheduling employed in SD; third, both the complexities and sizes of the data and model in EDM are much smaller than those in SD.

To address these challenges, we explore the integration of CFG and SiD for T2I diffusion distillation. In addition to testing the conventional approach of enhancing CFG on the pretrained score network, we introduce a novel strategy of reducing CFG on the fake score network, as well as a combined approach that employs both strategies, which we refer to as long and short guidance (LSG). This new method efficiently distills SD models into one-step T2I generators without requiring training data. Surprisingly, it achieves a new benchmark on the COCO-2014 validation set with a zero-shot FID score of **8.15**, the lowest to date for one-step, data-free distillation, despite not relying on additional regression or adversarial losses, nor real data—key components of recent distillation techniques.

The development of our SiD-LSG approach builds significantly on prior work in generative models, CFG, and acceleration methods. A comprehensive review of related work is provided in Appendix B.

# 2 DATA-FREE GUIDED SCORE IDENTITY DISTILLATION

We explore the use of SiD and CFG to distill SD, the leading open-source platform for T2I diffusion models that operate on the latent space of an image encoder-decoder (Rombach et al., 2022), with a specific focus on the data-free setting. Our focus is specifically on SD-v1-5 (SD1.5) and SD-v2-1-base (SD2.1-base), which are two versions extensively benchmarked for diffusion distillation.

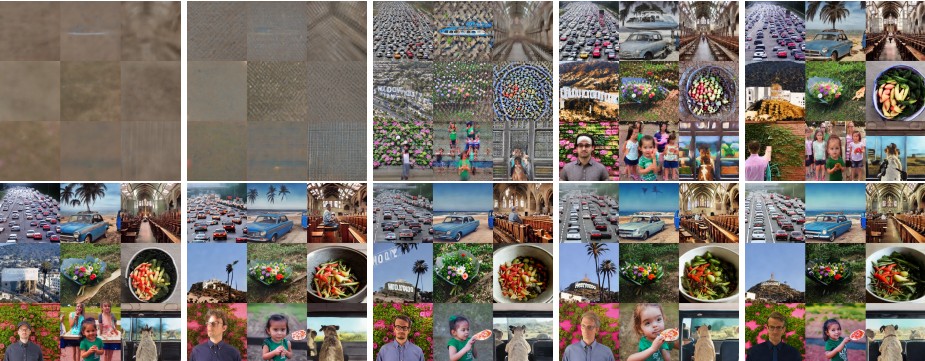

Figure 2: Rapid advancements in distilling Stable Diffusion 1.5 are showcased by the proposed SiD method that incorporates long-short guidance (LSG). Key parameters include a batch size of 512, a learning rate of 1e-6, and an LSG scale of 2. This data-free approach achieves a zero-shot FID of **9.56** on the COCO-2014 validation set, along with a competitive CLIP score of 0.313. By reducing the LSG scale to 1.5, the FID can be further lowered to a record **8.15** among data-free diffusion distillation models, with a corresponding CLIP score of 0.304. The series of images, generated from the same set of random noises post-training the SiD generator with varying counts of synthesized images, illustrates progressions at 0, 0.02, 0.1, 0.2, 0.5, 1, 2, 3, 4, and 5 million images. These are equivalent to 0, 40, 200, 400, 1k, 2k, 4K, 6K, 8k, and 10k training iterations respectively, organized from the top left to the bottom right. The progression of FIDs and CLIPs is detailed in the orange solid curves in the left plot of Fig. 4. The corresponding COCO-2014 validation text prompts are listed in Appendix F.

A significant hurdle involves incorporating CFG (Ho & Salimans, 2022), essential for T2I diffusion models to enhance their photorealism and text alignment, into the SiD loss functions. The second challenge is adapting SiD, initially tested with EDM noise scheduling, to the DDPM scheduling employed by SD. Addressing these challenges primarily requires modifying the derivation and loss functions of SiD. A third challenge arises because SD models are significantly larger and trained on bigger, more complex, and higher resolution data. Overcoming this challenge requires addressing numerous technical details, such as establishing the minimum hardware requirements and configuring the appropriate software settings to align with the constraints of the available computing platforms.

## 2.1 PRELIMINARIES ON SCORE IDENTITY DISTILLATION

We denote $c$ as the text representation of a pretrained text encoder, such as CLIP (Radford et al., 2021). Our objective is to distill a student model $p_\theta(x_g \mid c)$ from a pretrained T2I diffusion model, such as SD1.5, which can generate text-guided random samples in a single step as: $x_g = G_\theta(z, c)$, $z \sim p(z)$, where $G_\theta$ is a neural network parameterized by $\theta$ that deterministically transforms noise $z \sim p(z)$ into generated data $x_g$ under the guidance of text $c$. The distribution of $x_g$ is often implicit (Mohamed & Lakshminarayanan, 2016; Tran et al., 2017; Yin & Zhou, 2018), lacking an analytic probability density function (PDF) but is straightforward to sample from. The marginals of the real and generated data under the forward diffusion process can be expressed as:

$$p_{\text{data}}(x_t \mid c) = \int q(x_t \mid x_0) p_{\text{data}}(x_0 \mid c) \, dx_0, \quad p_\theta(x_t \mid c) = \int q(x_t \mid x_g) p_\theta(x_g \mid c) \, dx_g.$$

This structure, characterized by explicit conditional layers but implicit marginals, exemplifies a semi-implicit distribution (Yin & Zhou, 2018; Yu et al., 2023). This concept is employed by Zhou et al. (2024) to develop SiD, a method whose single-step data-free distillation capability has so far been demonstrated only on non-T2I diffusion models based on EDM.

We define the forward diffusion transition as $q(x_t \mid x_0) = \mathcal{N}(a_t x_0, \sigma_t^2 \mathbf{I})$, and unlike in SiD, which adheres to EDM noise scheduling where $a_t = 1$, we allow $a_t$ to vary within $[0, 1]$ to align with the DDPM scheduling used by SD. This necessitates generalizing the equations used in SiD by permitting $a_t \neq 1$. We note that other diffusion types, such as categorical (Austin et al., 2021; Hoogeboom et al., 2021; Gu et al., 2022; Hu et al., 2022), Poisson (Chen & Zhou, 2023), and beta diffusions (Zhou et al., 2023), also align with the semi-implicit framework and can potentially be adapted similarly.

**Score identities.** The scores $S(x_t) := \nabla_{x_t} \ln p_{\text{data}}(x_t \mid c)$ and $\nabla_{x_t} \ln p_\theta(x_t \mid c)$ are generally unknown. However, the score of the forward conditional $q(x_t \mid x) \sim \mathcal{N}(x, \sigma_t^2 \mathbf{I})$ is analytic:

$$\text{If } x_t = a_t x + \sigma_t \epsilon_t, \ \epsilon_t \sim \mathcal{N}(0, \mathbf{I}), \quad \text{then } \nabla_{x_t} \ln q(x_t \mid x) = \sigma_t^{-2}(a_t x - x_t) = -\sigma_t^{-1} \epsilon_t.$$

Exploiting the semi-implicit constructions, we follow SiD to present the following three identities:

$$\mathbb{E}[\boldsymbol{x}_0 \,|\, \boldsymbol{x}_t, \boldsymbol{c}] = \int \boldsymbol{x}_0 q(\boldsymbol{x}_0 \,|\, \boldsymbol{x}_t, \boldsymbol{c}) \, \mathrm{d}\boldsymbol{x}_0 = (\boldsymbol{x}_t + \sigma_t^2 \nabla_{\boldsymbol{x}_t} \ln p_{\mathrm{data}}(\boldsymbol{x}_t \,|\, \boldsymbol{c}))/a_t,$$

$$\mathbb{E}[\boldsymbol{x}_g \,|\, \boldsymbol{x}_t, \boldsymbol{c}] = \int \boldsymbol{x}_g q(\boldsymbol{x}_g \,|\, \boldsymbol{x}_t) \, \mathrm{d}\boldsymbol{x}_g = (\boldsymbol{x}_t + \sigma_t^2 \nabla_{\boldsymbol{x}_t} \ln p_\theta(\boldsymbol{x}_t \,|\, \boldsymbol{c}))/a_t,$$

$$\mathbb{E}_{p_\theta(\boldsymbol{x}_t \,|\, \boldsymbol{c})} \left[ u^T(\boldsymbol{x}_t) \nabla_{\boldsymbol{x}_t} \ln p_\theta(\boldsymbol{x}_t \,|\, \boldsymbol{c}) \right] = \mathbb{E}_{q(\boldsymbol{x}_t \,|\, \boldsymbol{x}_g) p_\theta(\boldsymbol{x}_g \,|\, \boldsymbol{c})} \left[ u^T(\boldsymbol{x}_t) \nabla_{\boldsymbol{x}_t} \ln q(\boldsymbol{x}_t \,|\, \boldsymbol{x}_g, \boldsymbol{c}) \right].$$

**MESM loss.** A pretrained T2I diffusion model, such as SD, provides a score network $S_\phi$ parameterized by $\phi$ that estimates the true data score as

$$-\sigma_t \nabla_{\boldsymbol{x}_t} \ln p_{\mathrm{data}}(\boldsymbol{x}_t \,|\, \boldsymbol{c}) \approx -\sigma_t S_\phi(\boldsymbol{x}_t, \boldsymbol{c}) := \sigma_t^{-1}(\boldsymbol{x}_t - a_t f_\phi(\boldsymbol{x}_t, t, \boldsymbol{c})) = \boldsymbol{\epsilon}_\phi(\boldsymbol{x}_t, \boldsymbol{c}).$$

It adopts $f_\phi(\boldsymbol{x}_t, t, \boldsymbol{c})$ as the functional approximation of the conditional expectation of the real image $\boldsymbol{x}_0$ given noisy image $\boldsymbol{x}_t$ and text $\boldsymbol{c}$, expressed as $\mathbb{E}[\boldsymbol{x}_0 \,|\, \boldsymbol{x}_t, \boldsymbol{c}]$, adopts $\boldsymbol{\epsilon}_\phi(\boldsymbol{x}_t, t, \boldsymbol{c})$ to predict the noise inside $\boldsymbol{x}_t$, and adopts $-\sigma_t^{-1} \boldsymbol{\epsilon}_\phi(\boldsymbol{x}_t, \boldsymbol{c})$ as the functional approximation of the true score $\nabla_{\boldsymbol{x}_t} \ln p_{\mathrm{data}}(\boldsymbol{x}_t \,|\, \boldsymbol{c})$. Given time step $t \sim p(t)$ and text $\boldsymbol{c}$, we define the model-based explicit score-matching (MESM) distillation loss, which is a form of Fisher divergence (Lyu, 2009; Holmes & Walker, 2017; Yang et al., 2019; Yu & Zhang, 2023), as

$$\mathcal{L}_\theta = \mathbb{E}_{\boldsymbol{x}_t \sim p_\theta(\boldsymbol{x}_t)}[\|S_\phi(\boldsymbol{x}_t, \boldsymbol{c}) - \nabla_{\boldsymbol{x}_t} \ln p_\theta(\boldsymbol{x}_t \,|\, \boldsymbol{c})\|_2^2]. \tag{1}$$

**Loss approximation for distillation.** The MESM loss in (1) is in general intractable to compute as $\nabla_{\boldsymbol{x}_t} \ln p_\theta(\boldsymbol{x}_t \,|\, \boldsymbol{c})$ is unknown. To denoise the noisy fake data $\boldsymbol{x}_t$ generated as

$$\boldsymbol{x}_t = a_t \boldsymbol{x}_g + \sigma_t \boldsymbol{\epsilon}_t, \;\; \boldsymbol{\epsilon}_t \sim \mathcal{N}(0, \mathbf{I}), \;\; \boldsymbol{x}_g = G_\theta(\boldsymbol{z}, \boldsymbol{c}), \;\; \boldsymbol{z} \sim p(\boldsymbol{z}), \tag{2}$$

there exists an optimal denoising network defined as $f_{\psi^*(\theta)}(\boldsymbol{x}_t, t, \boldsymbol{c}) = \mathbb{E}[\boldsymbol{x}_g \,|\, \boldsymbol{x}_t, \boldsymbol{c}] = (\boldsymbol{x}_t + \sigma_t^2 \nabla_{\boldsymbol{x}_t} \ln p_\theta(\boldsymbol{x}_t \,|\, \boldsymbol{c}))/a_t$. Given this optimal denoising network, the MESM loss in (1) would become

$$\mathcal{L}_\theta = \mathbb{E}_{\boldsymbol{x}_t \sim p_\theta(\boldsymbol{x}_t)}[\|a_t \sigma_t^{-2}(f_\phi(\boldsymbol{x}_t, t, \boldsymbol{c}) - f_{\psi^*(\theta)}(\boldsymbol{x}_t, t, \boldsymbol{c})\|_2^2]. \tag{3}$$

As $\psi^*(\theta)$ is unknown in practice, we follow SiD to alternates between optimizing $\psi$ and $\theta$ using

$$\min_\psi \hat{\mathcal{L}}_\psi(\boldsymbol{x}_t, c, t) = \frac{a_t^2}{\sigma_t^2}\|f_\psi(\boldsymbol{x}_t, t, \boldsymbol{c}) - \boldsymbol{x}_g\|_2^2 = \|\boldsymbol{\epsilon}_\psi(\boldsymbol{x}_t, t, \boldsymbol{c}) - \boldsymbol{\epsilon}_t\|_2^2, \tag{4}$$

$$\min_\theta \tilde{L}_\theta(\boldsymbol{x}_t, t, \phi, \psi) = \omega(t)\frac{a_t^2}{\sigma_t^4}(f_\phi(\boldsymbol{x}_t, t, \boldsymbol{c}) - f_\psi(\boldsymbol{x}_t, t, \boldsymbol{c}))^T(f_\psi(\boldsymbol{x}_t, t, \boldsymbol{c}) - \boldsymbol{x}_g), \tag{5}$$

$$= \omega(t)\frac{1}{\sigma_t^2}(\boldsymbol{\epsilon}_\psi(\boldsymbol{x}_t, t, \boldsymbol{c}) - \boldsymbol{\epsilon}_\phi(\boldsymbol{x}_t, t, \boldsymbol{c}))^T(\boldsymbol{\epsilon}_t - \boldsymbol{\epsilon}_\psi(\boldsymbol{x}_t, t, \boldsymbol{c})),$$

where $\boldsymbol{x}_t$ is generated as in (2) and $\omega(t)$ are weighted coefficients that will be specified.

## 2.2 SiD with classifier-free guidance

An essential practice for enhancing photorealism and alignment with text instructions in T2I diffusion models involves incorporating CFG into reverse diffusion. This principle also applies to distillation methods for these models, where CFG must be integrated into the appropriate terms of their distillation loss functions. Therefore, a key distinction in the distillation of SD, compared to previous unconditional and label-conditional diffusion models, lies in the need to introduce CFG.

SiD presents a unique opportunity to apply CFG to enhance its T2I generation performance. First, we note CFG enhances text guidance by modifying the distribution of $\boldsymbol{x}_t$ given text $\boldsymbol{c}$ as

$$p(\boldsymbol{x}_t \,|\, \boldsymbol{c}, \kappa) \propto p(\boldsymbol{c} \,|\, \boldsymbol{x}_t)^\kappa p(\boldsymbol{x}_t) \propto \left(\tfrac{p(\boldsymbol{x}_t \,|\, \boldsymbol{c})}{p(\boldsymbol{x}_t)}\right)^\kappa p(\boldsymbol{x}_t),$$

which means $\nabla_{\boldsymbol{x}_t} \ln p(\boldsymbol{x}_t \,|\, \boldsymbol{c}, \kappa) = \nabla_{\boldsymbol{x}_t} \ln p(\boldsymbol{x}_t) + \kappa[\nabla_{\boldsymbol{x}_t} \ln p(\boldsymbol{x}_t \,|\, \boldsymbol{c}) - \nabla_{\boldsymbol{x}_t} \ln p(\boldsymbol{x}_t)]$. Second, the score network, when reaching its optimal, is related to the true score as $f(\boldsymbol{x}_t, t, \boldsymbol{c}) = a_t^{-1}(\boldsymbol{x}_t + \sigma_t^2 \nabla_{\boldsymbol{x}_t} \ln p(\boldsymbol{x}_t \,|\, c)) = a_t^{-1}(\boldsymbol{x}_t - \sigma_t \boldsymbol{\epsilon}(\boldsymbol{x}_t, t, \boldsymbol{c}))$. Therefore, with "$\cdot$" representing $\phi$ or $\psi$, we can express the score network $f.(\boldsymbol{x}_t, t, \boldsymbol{c})$ under CFG with scale $\kappa$ as

$$f_{\cdot,\kappa}(\boldsymbol{x}_t, t, \boldsymbol{c}) = f.(\boldsymbol{x}_t, t) + \kappa[f.(\boldsymbol{x}_t, t, \boldsymbol{c}) - f.(\boldsymbol{x}_t, t)]. \tag{6}$$

Similarly, for noise prediction network, we have $\boldsymbol{\epsilon}_{\cdot,\kappa}(\boldsymbol{x}_t, t, \boldsymbol{c}) = \boldsymbol{\epsilon}.(\boldsymbol{x}_t, t) + \kappa[\boldsymbol{\epsilon}.(\boldsymbol{x}_t, t, c) - \boldsymbol{\epsilon}.(\boldsymbol{x}_t, t)]$. With CFG, the score and noise prediction networks are related in the same way, which means that $\boldsymbol{\epsilon}_{\cdot,\kappa}(\boldsymbol{x}_t, t, \boldsymbol{c}) = \sigma_t^{-1}(\boldsymbol{x}_t - a_t f_{\cdot,\kappa}(\boldsymbol{x}_t, t, \boldsymbol{c}))$ and $f_{\cdot,\kappa}(\boldsymbol{x}_t, t, \boldsymbol{c}) = a_t^{-1}(\boldsymbol{x}_t - \sigma_t \boldsymbol{\epsilon}_{\cdot,\kappa}(\boldsymbol{x}_t, t, \boldsymbol{c}))$.

Inspecting the two losses in (4) and (5) suggests four potential places to inject CFG. More specifically, with $\kappa_1, \kappa_2, \kappa_3, \kappa_4 \in \mathbb{R}_+$, where $\mathbb{R}_+ = \{x : x \geq 0\}$, we modify the losses with CFGs as

$$\hat{\mathcal{L}}_\psi(\boldsymbol{x}_t, c, t) = \frac{a_t^2}{\sigma_t^2} \| f_{\psi,\kappa_1}(\boldsymbol{x}_t, t, \boldsymbol{c}) - \boldsymbol{x}_g \|_2^2, = \| \boldsymbol{\epsilon}_{\psi,\kappa_1}(\boldsymbol{x}_t, t, \boldsymbol{c}) - \boldsymbol{\epsilon}_t \|_2^2, \tag{7}$$

$$\tilde{L}_\theta(\boldsymbol{x}_t, t, \phi, \psi) = \omega(t) \frac{a_t^4}{\sigma_t^4} (f_{\phi,\kappa_4}(\boldsymbol{x}_t, t, \boldsymbol{c}) - f_{\psi,\kappa_2}(\boldsymbol{x}_t, t, \boldsymbol{c}))^T (f_{\psi,\kappa_3}(\boldsymbol{x}_t, t, \boldsymbol{c}) - \boldsymbol{x}_g). \tag{8}$$

## 2.3 LONG AND SHORT GUIDANCE

Previous works equipped with a fake score network $f_\psi$ typically only consider adding CFG when evaluating the pretrained score network $f_\phi$, such as in DMD (Yin et al., 2023). In the context of SiD, this corresponds to setting $\kappa_1 = \kappa_2 = \kappa_3 = 1$ and $\kappa_4 > 1$. In this paper, we discover that a broad spectrum of combinations of $\kappa_1$, $\kappa_2$, $\kappa_3$, and $\kappa_4$ can all significantly enhance performance compared to not using any CFG at all, which means setting $\kappa_1 = \kappa_2 = \kappa_3 = \kappa_4 = 1$. These different combinations are found to lead to different balances of minimizing FID, which reflects how well the generated data match the training data in distribution, and maximizing the CLIP score (Radford et al., 2021), which reflects how well the generated images follow the textual guidance. This flexibility to accommodate various CFG combinations expands the design space for SD distillation, balancing generation quality and text adherence. However, given the vastness of the search space, we are motivated to develop strategies that constrain the scope of exploration. We present three such strategies, acknowledging that there may be more effective approaches not explored in this paper.

**Long strategy: Enhancing CFG of the pretrained score network $f_\phi$.** Aligning with established practices, the most common approach involves enhancing the CFG of the pretrained score network $f_\phi$. An example setting under this strategy in SiD is: $\kappa_1 = \kappa_2 = \kappa_3 = 1$ and $\kappa_4 = 3$. The rationale is that by biasing the teacher $f_\phi$ to favor generations more aligned with $\boldsymbol{c}$ using a CFG scale greater than 1, such as $\kappa_4 = 3$, the student generator is compelled to follow suit. We will present experimental results to demonstrate the effectiveness of this strategy while also discussing its limitations.

**Short strategy: Weakening CFG of the fake score network $f_\psi$.** With the availability of a fake score network $f_\psi$, we have developed a new strategy to enhance SiD's T2I generation capabilities by reducing the CFG of $f_\psi$ in SiD. An exemplary configuration is setting $\kappa_1 = \kappa_4 = 1$ while allowing $\kappa_2 = \kappa_3$ to vary within $(0, 1)$. The underlying idea is that by diminishing $f_\psi$'s capacity to detect generations aligned with $\boldsymbol{c}$ through a reduced CFG scale ($0 < \kappa_2 < 1$), the student generator is incentivized to produce images that better align with the textual guidance to compensate for this reduction. We will present experiments to assess the efficacy and limitations of this strategy.

**Long and short CFGs.** We introduce an innovative strategy termed long-short guidance (LSG), where we enhance CFG during the training of $f_\psi$ by setting $\kappa_1 > 1$, and maintain CFGs on $f_\psi$ and $f_\phi$ during the training of $G_\theta$ at the same or a lower scale by setting $1 \leq \kappa_2 = \kappa_3 = \kappa_4 \leq \kappa_1$. The logic behind LSG is that enhancing the CFG of $f_\psi$ during training effectively reduces its CFG at evaluation time when used at the same or a reduced level. To our knowledge, we are the first to incorporate CFG into the training of the fake score network $f_\psi$. LSG has been shown to effectively balance reducing FID scores while increasing CLIP scores, and will therefore be used as the default method.

We refer the reader to Appendix C for further discussion on the long and short CFG strategies.

## 3 EXPERIMENTS

We summarize the parameter settings, such as batch size, learning rate, and optimizer configurations, in Table 4 in Appendix D. Unless specified in ablation studies, the settings are uniformly applied across all guidance strategies. We present the details of our method in Algorithm 1, where we generate an image in one step and generalize the setting in DMD and SiD to set $\omega(t)$ as

$$\boldsymbol{x}_g = G_\theta(\boldsymbol{z}, \boldsymbol{c}) = f_\theta(\sigma_{t_{init}} \boldsymbol{z}, t_{init}, \boldsymbol{c}), \quad \boldsymbol{z} \sim \mathcal{N}(0, \mathbf{I}); \quad \omega(t) = \frac{\sigma_t^4}{a_t^2} \frac{C}{\| \boldsymbol{x}_g - f_{\phi,\kappa_4}(\boldsymbol{x}_t, t, \boldsymbol{c}) \|_{1,sg}}, \tag{9}$$

where $C$ is the total number of pixels in $\boldsymbol{x}_g$. We initialize both $\psi$ and $\theta$ using the pretrained $\phi$ from SD. We have tested $t_{init} \in \{354, 550, 625, 675, 800, 900, 999\}$ and found that the model's performance under FP32 optimization is not sensitive to these choices. To align with the setting in SiD, we set $t_{init} = 625$, which would result in $\sigma_{t_{init}}/a_{t_{init}} = 2.5$ under the DDPM schedule used to train SD1.5.

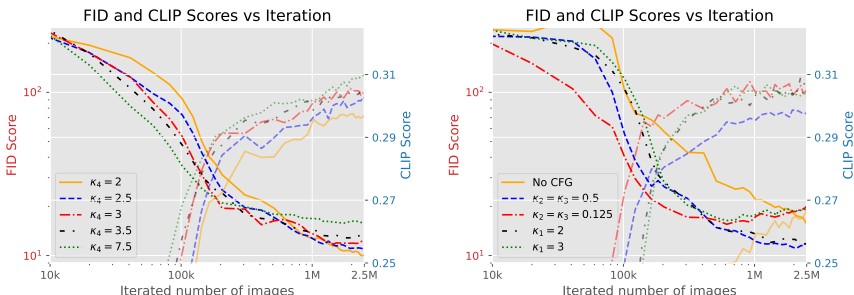

Figure 3: **Left (Long CFG of the true score network):** This plot illustrates the gradual decline in FID and the corresponding rise in CLIP scores, each influenced by different CFGs applied to the true score network. $\kappa$ values not specified in the legend are set to 1. FID scores are plotted on the primary y-axis, while CLIP scores are displayed on the secondary y-axis in corresponding line styles but with slight transparency. Together, these lines demonstrate how various CFGs impact model performance. **Right (No CFG; Short CFG of the fake score network with $\kappa_2 = \kappa_3 \in (0, 1)$; a simple form of LSG that sets $\kappa_1 > 1$):** Analogous plot to the left where the CFGs of the fake score network are not applied, weakened during evaluation, or enhanced during training.

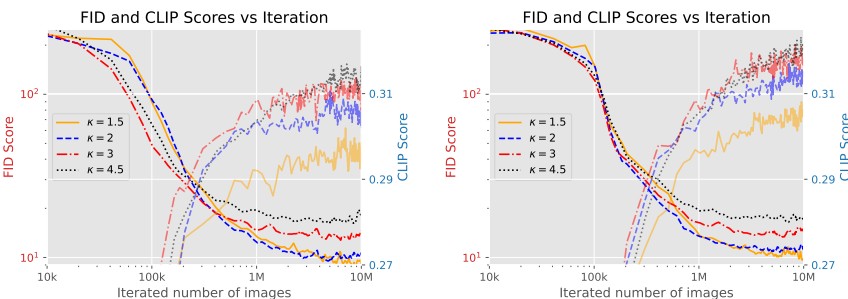

Figure 4: The plot with the proposed long-short guidance (LSG) demonstrates the FID and CLIP progressions of SiD in SD1.5 (left panel) and SD2.1-base (right panel).

We conduct a comprehensive study to evaluate the performance of SiD using the proposed LSG for distilling SD1.5. Additionally, we apply the same LSG scales to distill SD2.1-base, further assessing the adaptability and effectiveness of our approach across different model versions.

We consider a standard setting that utilizes the Aesthetics6+ prompt (Cherti et al., 2023) for training and evaluates performance by computing zero-shot FID on the COCO-2014 validation set. We adhere to the standard protocol by generating 30k images to compare with the 40,504 images in the COCO-2014 validation set for calculating zero-shot FID. Additionally, we employ the `ViT-g-14-laion2b_s12b_b42k` encoder (Ilharco et al., 2021; Cherti et al., 2023) to compute the CLIP score (Radford et al., 2021). The FID and CLIP scores presented in the figures are calculated using randomly sampled prompts from the COCO-2014 validation set. When reporting the FID and CLIP results of SiD in Table 1, we use the exact evaluation code[1] provided by GigaGAN (Kang et al., 2023), where a pre-defined list of 30k text prompts selected from the COCO-2014 validation set is used to generate the 30k images, which are used for computing FID and CLIP scores with images from the validation set as the reference.

## 3.1 LONG AND/OR SHORT GUIDANCE STRATEGIES

**No CFG.** Figure 3 demonstrates that without CFG, where $\kappa_1 = \kappa_2 = \kappa_3 = \kappa_4 = 1$, results are underwhelming, highlighting the need for CFG in SiD to distill T2I diffusion models.

**Long the CFG of the pretrained score network.** For the long strategy, we set $\kappa_1 = \kappa_2 = \kappa_3 = 1$ and $\kappa_4 > 1$, specifically exploring values of $\kappa_4$ from the set $\{2, 2.5, 3, 3.5, 7.5\}$. As shown in the left panel of Figure 3, our experiments results indicate that this setup yields highly competitive performance, even after processing as few as 2.5 million fake images (approximately 5,000 iterations with a mini-batch size of 512). The parameter $\kappa_4$ plays a crucial role in balancing FID and CLIP scores. For achieving lower FID scores, $\kappa_4 = 2$ can yield an FID close to 10, while for higher CLIP

---

[1]https://github.com/mingukkang/GigaGAN/tree/main/evaluation

Table 1: Comparison of image generation methods across various metrics. Inference times are estimated using a Nvidia A100 GPU as reference. The numbers of the methods marked with $^{\dagger}$ are produced by running the publicly available model checkpoints. Other data are sourced from corresponding scientific papers with comparisons as reported. The value followed by symbol $\sim$ indicates it is estimated based on the plots shown in the paper.

| Method | Res. | Time ($\downarrow$) | # Steps | # Param. | FID ($\downarrow$) | CLIP ($\uparrow$) |
|---|---|---|---|---|---|---|
| Autoregressive Models | | | | | | |
| DALL·E (Ramesh et al., 2021) | 256 | - | - | 12B | 27.5 | - |
| CogView2 (Ding et al., 2021) | 256 | - | - | 6B | 24.0 | - |
| Parti-750M (Yu et al., 2022) | 256 | - | - | 750M | 10.71 | - |
| Parti-3B (Yu et al., 2022) | 256 | 6.4s | - | 3B | 8.10 | - |
| Parti-20B (Yu et al., 2022) | 256 | - | - | 20B | 7.23 | - |
| Make-A-Scene (Gafni et al., 2022) | 256 | 25.0s | - | - | 11.84 | - |
| Masked Models | | | | | | |
| Muse (Chang et al., 2023) | 256 | 1.3 | 24 | 3B | 7.88 | 0.32 |
| Diffusion Models | | | | | | |
| GLIDE (Nichol et al., 2022) | 256 | 15.0s | 250+27 | 5B | 12.24 | - |
| DALL·E 2 (Ramesh et al., 2022) | 256 | - | 250+27 | 5.5B | 10.39 | - |
| LDM-KL-8-G* (Rombach et al., 2022) | 256 | 3.7s | 250 | 1.45B | 12.63 | - |
| Imagen (Ho et al., 2022) | 256 | 9.1s | - | 3B | 7.27 | - |
| eDiff-I (Balaji et al., 2022) | 256 | 32.0s | 25+10 | 9B | 6.95 | - |
| Generative Adversarial Networks (GANs) | | | | | | |
| LAFITE (Zhou et al., 2022) | 256 | 0.02s | 1 | 75M | 26.94 | - |
| StyleGAN-T (Sauer et al., 2023a) | 512 | 0.10s | 1 | 1B | 13.90 | $\sim$0.293 |
| GigaGAN (Kang et al., 2023) | 512 | 0.13s | 1 | 1B | 9.09 | - |
| Distilled Stable Diffusion 2.1 | | | | | | |
| $^{\dagger}$ADD (SD-Turbo) (Sauer et al., 2023b) | 512 | - | 1 | - | 16.25 | 0.335 |
| Distilled Stable Diffusion XL | | | | | | |
| $^{\dagger}$ADD (SDXL-Turbo) (Sauer et al., 2023b) | 512 | - | 1 | - | 19.08 | 0.343 |
| Stable Diffusion 1.5 and its accelerated or distilled versions | | | | | | |
| SD1.5 (CFG=3) (Rombach et al., 2022) | 512 | 2.59s | 50 | 0.9B | 8.78 | - |
| SD1.5 (CFG=8) (Rombach et al., 2022) | 512 | 2.59s | 50 | 0.9B | 13.45 | 0.322 |
| DPM++ (4 step) (Lu et al., 2022a) | 512 | 0.26s | 4 | 0.9B | 22.44 | 0.31 |
| UniPC (4 step) (Zhao et al., 2023) | 512 | 0.26s | 4 | 0.9B | 22.30 | 0.31 |
| LCM-LoRA (4 step) (Luo et al., 2023b) | 512 | 0.19s | 4 | 0.9B | 23.62 | 0.30 |
| LCM-LoRA (1 step) (Luo et al., 2023b) | 512 | 0.07s | 1 | 0.9B | 77.90 | 0.24 |
| InstaFlow-0.9B (Liu et al., 2023) | 512 | 0.09s | 1 | 0.9B | 13.10 | 0.28 |
| InstaFlow-1.7B (Liu et al., 2023) | 512 | 0.12s | 1 | 1.7B | 11.83 | - |
| UFOGen (Xu et al., 2023) | 512 | 0.09s | 1 | 0.9B | 12.78 | - |
| DMD (CFG=3) (Yin et al., 2023) | 512 | 0.09s | 1 | 0.9B | 11.49 | - |
| DMD (CFG=8) (Yin et al., 2023) | 512 | 0.09s | 1 | 0.9B | 14.93 | 0.32 |
| BOOT (Gu et al., 2023) | 512 | 0.09s | 1 | 0.9B | 48.20 | 0.26 |
| Guided Distillation (Meng et al., 2023) | 512 | - | 1 | 0.9B | 37.3 | 0.27 |
| SiD-LSG ($\kappa = 1.5$) | 512 | 0.09s | 1 | 0.9B | 8.71 | 0.302 |
| SiD-LSG ($\kappa = 1.5$, double the training time) | 512 | 0.09s | 1 | 0.9B | **8.15** | 0.304 |
| SiD-LSG ($\kappa = 2$) | 512 | 0.09s | 1 | 0.9B | 9.56 | 0.313 |
| SiD-LSG ($\kappa = 3$) | 512 | 0.09s | 1 | 0.9B | 13.21 | 0.314 |
| SiD-LSG ($\kappa = 4.5$) | 512 | 0.09s | 1 | 0.9B | 16.59 | **0.317** |
| Stable Diffusion 2.1-base and its distilled versions | | | | | | |
| SD2.1-base (Rombach et al., 2022) | 512 | 0.09s | 1 | 0.9B | 202.14 | 0.08 |
| SD2.1 base (Rombach et al., 2022) | 512 | 0.77s | 25 | 0.9B | 13.45 | 0.30 |
| SwiftBrush (Nguyen & Tran, 2024) | 512 | 0.09s | 1 | 0.9B | 16.67 | 0.29 |
| SiD-LSG ($\kappa = 1.5$) | 512 | 0.09s | 1 | 0.9B | **9.52** | 0.308 |
| SiD-LSG ($\kappa = 2$) | 512 | 0.09s | 1 | 0.9B | 10.97 | 0.318 |
| SiD-LSG ($\kappa = 3$) | 512 | 0.09s | 1 | 0.9B | 13.50 | 0.321 |
| SiD-LSG ($\kappa = 4.5$) | 512 | 0.09s | 1 | 0.9B | 16.54 | **0.322** |

scores, $\kappa_4 = 7.5$ can result in a CLIP score around 0.31. However, our primary objective is to devise a strategy that lowers FID while minimizing CLIP degradation.

**Short the CFG of the fake score network.** For the short strategy, we set $\kappa_1 = \kappa_4 = 1$ and explore $\kappa_2 = \kappa_3 \in \{0.5, 0.125\}$. As illustrated in the right panel of Figure 3, this approach delivers competitive performance, with $\kappa_2$ dictating the balance between FID and CLIP scores. However, compared to the long strategy, this configuration generally produces inferior results, as suggested by lower CLIP scores when FIDs are controlled to similar levels.

**LSG: Long and Short classifier-free Guidance.** Below, we explore how to effectively integrate the long and short strategies to enhance text guidance in diffusion distillation. Initially, we discovered a "simplest" form of LSG strategy by amplifying the CFG during the training of $f_\psi$. Specifically,

Table 2: Comparison of HPSv2 score and Precision/Recall on the COCO-2014 validation set. The HPSv2 scores of ADD (SD-Turbo) are produced based on the publicly available model checkpoint. The HPSv2 scores of the other baselines are quoted from SwiftBrush (Nguyen & Tran, 2024). The Precision and Recall on COCO-2014 are obtained using the 30K images generated by the corresponding model checkpoints.

| Teacher | Student | Human Preference Score v2 ↑ | | | | COCO-2014 Precision & Recall ↑ | |
| | | Anime | Photo | Concept Art | Paintings | Precision | Recall |
|---|---|---|---|---|---|---|---|
| SD2.1 | ADD (SD-Turbo) (Sauer et al., 2023b) | **27.48** | 26.89 | **26.86** | 27.46 | 0.65 | 0.35 |
| SD1.5 | LCM (Luo et al., 2023b) | 22.61 | 22.71 | 22.74 | 22.91 | - | - |
| SD1.5 | InstaFlow (Liu et al., 2023) | 25.98 | 26.32 | 25.79 | 25.93 | 0.53 | 0.45 |
| SD1.5 | BOOT (Gu et al., 2023) | 25.29 | 25.16 | 24.40 | 24.61 | - | - |
| SD2.1-base | SwiftBrush (Nguyen & Tran, 2024) | 26.91 | 27.21 | 26.32 | 26.37 | 0.47 | 0.46 |
| SD1.5 | SiD-LSG ($\kappa = 1.5$) | 26.58 | 26.80 | 26.02 | 26.02 | 0.59 | 0.52 |
| | SiD-LSG ($\kappa = 1.5$, double the training time) | 26.58 | 26.80 | 26.01 | 26.02 | 0.60 | **0.53** |
| | SiD-LSG ($\kappa = 2$) | 26.94 | 27.03 | 26.35 | 26.27 | 0.64 | 0.48 |
| | SiD-LSG ($\kappa = 3$) | 27.10 | 27.11 | 26.47 | 26.46 | 0.65 | 0.40 |
| | SiD-LSG ($\kappa = 4.5$) | 27.39 | 27.30 | 26.65 | 26.58 | **0.67** | 0.34 |
| SD2.1-base | SiD-LSG ($\kappa = 1.5$) | 26.65 | 26.87 | 26.19 | 26.14 | 0.60 | 0.49 |
| | SiD-LSG ($\kappa = 2$) | 26.90 | 27.08 | 26.43 | 26.47 | 0.62 | 0.44 |
| | SiD-LSG ($\kappa = 3$) | 27.27 | 27.22 | 26.75 | 26.72 | 0.64 | 0.38 |
| | SiD-LSG ($\kappa = 4.5$) | 27.42 | **27.31** | 26.81 | 26.79 | 0.63 | 0.34 |

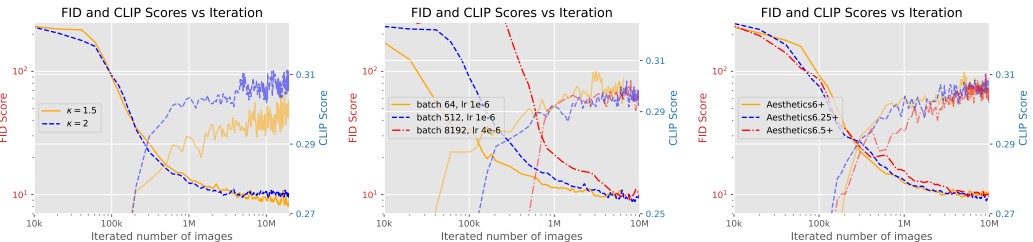

Figure 5: This figure illustrates the progression of FID and CLIP scores during an ablation study of distilling SD1.5 using SiD-LSG. The default settings of batch size 512, learning rate 1e-6, LSG scale 2, and Prompt Aesthetics6+ are maintained unless specified otherwise. **Left**: The number of training fake images is doubled from 10M to 20M under LSG scales of 1.5 and 2.0. **Middle**: Variations in batch size and learning rate settings under LSG 1.5. **Right**: Comparison of training prompts Aesthetics6+, Aesthetics6.25+, and Aesthetics6.5+.

we set $\kappa_1$ to 2 or 3, while keeping $\kappa_2 = \kappa_3 = \kappa_4$ at 1. As illustrated in the right panel of Figure 3, setting $\kappa_1 = 3$ (green lines) proves as effective as the short strategy with $\kappa_2 = \kappa_3 = 0.125$ (red lines), and $\kappa_1 = 2$ (black lines) outperforms the short strategy with $\kappa_2 = \kappa_3 = 0.5$ (blue lines) and is comparable to the long strategy with $\kappa_4 = 3$ (red lines in the left panel). These findings validate this "simplest" LSG as an effective guidance strategy in text-guided diffusion distillation.

Nevertheless, merely matching the performance of the best long or short strategy is insufficient to justify adopting this "simplest" LSG. The efficacy of LSG is notably enhanced when the CFG scale applied to the fake score network during its training exceeds 1 and is maintained throughout the generator's training. This strategy strikes an effective balance between minimizing FID and maximizing CLIP. Specifically, for this recommended LSG configuration, we evaluated $\kappa_1 = \kappa_2 = \kappa_3 = \kappa_4$ values within $\{1.5, 2, 3, 4.5\}$, presenting the results in Figure 4 for distilling both SD 1.5 and 2.1-base models. Comparisons between SD1.5 outcomes in the left panel of Figure 4 and those in Figure 3 demonstrate superior performance with this LSG setting, indicated by higher CLIP scores at controlled FID levels, and lower FID scores at controlled CLIP levels. Generally, within the range of 1.5 to 4.5, a lower guidance scale correlates with better FID but worse CLIP, and vice versa, as evidenced by the curves for both SD 1.5 and 2.1-base in Figure 4.

## 3.2 ABLATION STUDY

We investigate the impact of extended training durations under two different guidance scales, variations in batch size, and the selection of training prompts. Initially, we doubled the number of fake images used to train the generator from 10M to 20M and monitored the evolution of the FID and CLIP scores. From the left panel of Figure 5, we observe continuous improvements in both FID and CLIP scores for an LSG of 1.5 when training is extended beyond 10M fake images, and sustained enhancements in CLIP scores for an LSG of 2.0. Notably, by doubling the training input from 10M to 20M fake images, the FID under LSG 1.5 decreased from 8.71 to a record low of **8.15** among diffusion distillation methods in the data-free setting.

We assessed the effects of batch size by considering two additional settings: a batch size of 8192 with a learning rate of 4e-6, a configuration used in SiD to distill the EDM model pretrained on ImageNet 64x64 (Zhou et al., 2024), and a batch size of 64 with a learning rate of 1e-6. The middle panel of Figure 5 demonstrates that while there are initial differences in the convergence speed in terms of the number of fake images processed, the performances eventually converge to similar levels. We note that while smaller batch sizes may seem to converge faster, they require more time to process the same number of images. This is due to more frequent model parameter updates, higher communication costs between GPUs, and additional overheads typically associated with smaller batch sizes.

Lastly, we explored the effect of changing training prompts, shifting from the Aesthetics6+ prompts to Aesthetics6.25+ and Aesthetics6.5+ prompts (Cherti et al., 2023). The performance, as shown in the right panel of Figure 5, appears comparable across these variations. Specifically, under an LSG of 2.0, switching from Aesthetics6+ to Aesthetics6.25+ enabled us to further reduce the FID from the 9.56 reported in Table 1 to 9.21, although the CLIP score decreased slightly from 0.313 to 0.311, indicating no significant performance disparity between them.

The results of the ablation study show that SiD-LSG has low sensitivity to variations in batch size and training prompts and its performance could potentially be further enhanced with extended training.

### 3.3 QUANTITATIVE AND QUALITATIVE EVALUATIONS

We present comprehensive results from prior studies across various experimental settings, including both one-step and multi-step generation methods. When evaluation results are available in existing literature (Yin et al., 2023; Liu et al., 2023; Kang et al., 2023; Nguyen & Tran, 2024), we directly cite them; otherwise, if model checkpoints are accessible, either publicly or provided upon request by the authors, we utilize the evaluation code from GigaGAN to produce the reported results. For our SiD-LSG, we select $\kappa_1 = \kappa_2 = \kappa_3 = \kappa_4 \in \{1.5, 2.0, 3.0, 4.5\}$.

For comparisons of FID and CLIP scores, the results are detailed in Table 1. Among all one-step distillation methods, our approach notably excels in zero-shot text-conditioned image generation on the COCO-2014 dataset, as reflected by both FID-30K and CLIP scores. Specifically, with the guidance scale set as 2, our method attains FID scores as low as 9.56 and 10.97, and a CLIP score above 0.31 and around 0.32, using SD 1.5 and 2.1-base as the pretrained backbones, respectively. Notably, by setting the guidance scale to 1.5 and doubling the training time, our method achieves a record-low data-free FID of **8.15**, along with a CLIP score of 0.304, when distilling SD1.5. These results remain highly competitive when compared to other generative approaches, such as autoregressive models and GANs, and are even comparable to previous multi-step diffusion-based sampling methods. Analyzing different $\kappa$ values, we observe a trade-off between FID and CLIP scores: smaller $\kappa$ values generally yield better FID metrics, while larger values enhance CLIP scores, aligning with past findings on the impact of guidance scale.

Beyond FID and CLIP scores, we also assess Precision and Recall (Kynkäänniemi et al., 2019) as well as Human Preference Score (HPSv2) (Wu et al., 2023), which are presented in Table 2. We reuse the same 30k images from previous evaluations for the Precision and Recall calculations. For HPSv2, we follow their established protocol, generating images from 800 text prompts per category. Except for ADD trained with real data and adversarial loss, our SiD-LSG models outperform other baselines in HPSv2 scores across all categories, as well as in Precision and Recall metrics. Notably, with $\kappa = 4.5$, SiD-LSG reaches peak performance in distilling SD2.1-base. Regarding Precision and Recall, higher $\kappa$ values lead to improved Precision, while lower values result in better Recall.

For qualitative verification, we utilize our models with $\kappa = 4.5$, selecting six prompts from across all HPSv2 categories to generate images. To ensure a fair comparison, we maintain the same random seed for image generation across all methods. The visual results are illustrated in Figure 6, where SiD consistently shows superior text-image alignment and visual fidelity.

To contextualize the numerical differences in metrics such as FID, CLIP, and HPSv2, we present Figures 7 and 8 to illustrate their implications for visual perception: the model with the best FID excels in diversity, while the model with the best CLIP stands out in text alignment and aesthetic quality.

For broader impact, please refer to the discussion in Appendix A. For limitations and computational requirements, please refer to a detailed discussion in Appendix E.

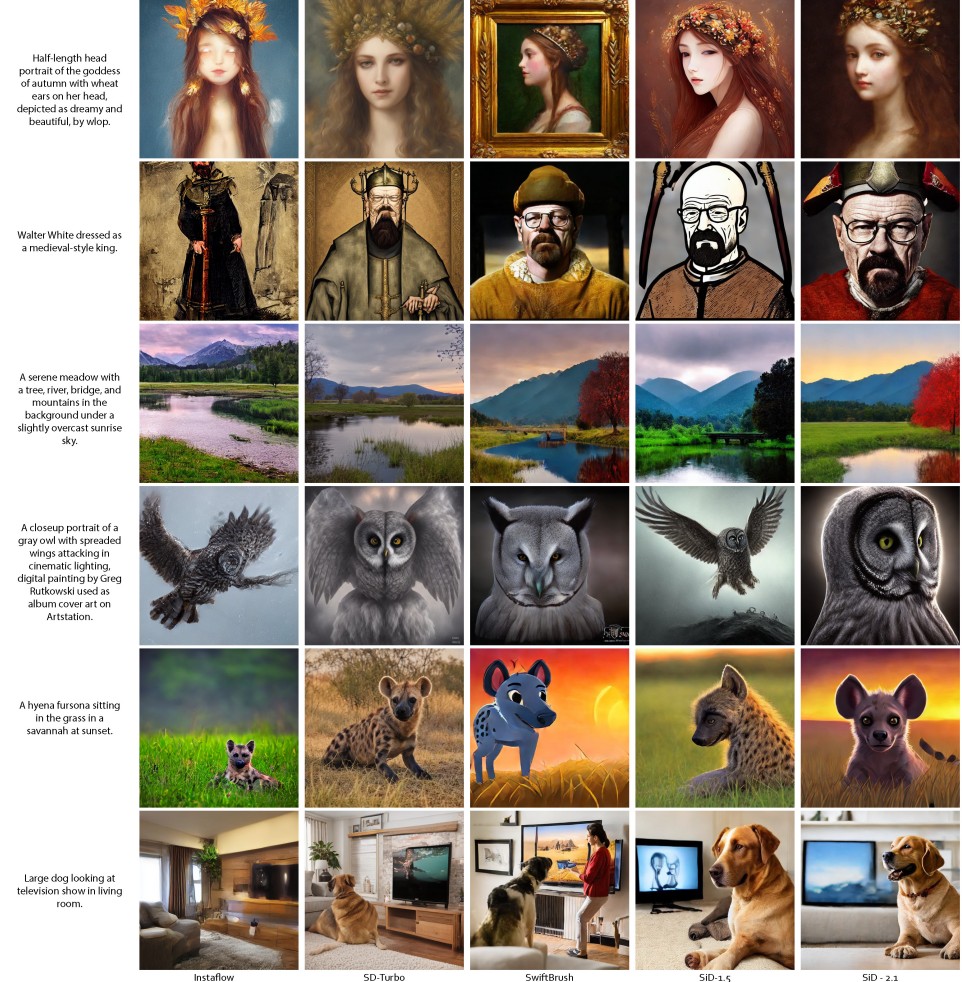

Figure 6: Qualitative comparison of one-step distillation methods using identical text prompts and random seeds.

## 4 CONCLUSION AND FUTURE WORK

This paper introduces a novel data-free method combining Classifier-Free Guidance (CFG) with Score identity Distillation (SiD) to efficiently distill Stable Diffusion models into effective one-step generators. By leveraging our innovative Long and Short CFG strategies (LSG), we distilled these models using only synthetic images generated by the one-step generator. This approach not only validates the practical potential of SiD but also sets new benchmarks for data-free one-step diffusion distillation, achieving remarkable zero-shot FID scores on the COCO-2014 validation set. Our method enhances efficiency while maintaining generation performance, allowing learning from the teacher model without the need for real images or the inclusion of additional regression or adversarial losses. We will make our code and distilled models publicly available to facilitate further research.

In the data-free setting, we are exploring the use of SiD-LSG for privacy and security-sensitive tasks where access to actual training data is not feasible. While we have advanced the capabilities of data-free diffusion distillation, our baseline methods such as ADD and DMD, typically require the use of real or teacher-synthesized images. Moving forward, we plan to lift the data-free constraint and integrate SiD-LSG with Diffusion GAN-based adversarial training, which has successfully transformed pretrained unconditional and label-conditional diffusion models into one-step generators, achieving state-of-the-art generation performance without CFG (Zhou et al., 2025). This transition will involve the use of real images to not only further enhance photo-realism and improve text alignment but also adapt SiD-LSG distilled one-step generators to domains that differ from those used to train the teacher. This approach aims to broaden the applicability and effectiveness of our SiD-LSG distilled diffusion models.

## ACKNOWLEDGMENTS

M. Zhou, Z. Wang, and H. Zheng acknowledge the support of NSF-IIS 2212418 and NIH-R37 CA271186.

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

# Guided Score Identity Distillation for Data-Free One-Step Text-to-Image Generation: Appendix

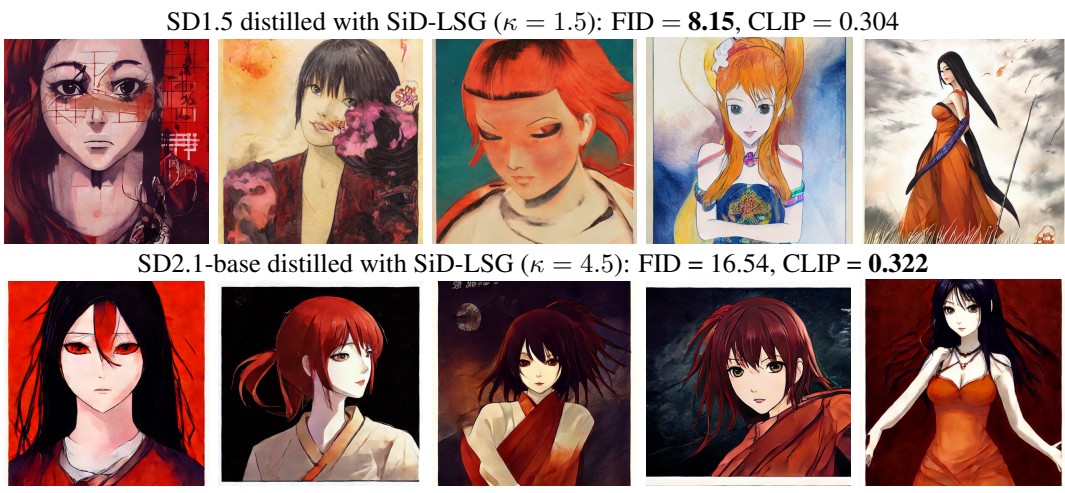

Figure 7: Visual comparison of two SiD-LSG models: one preferred for FID and the other for CLIP. All images are generated from the same text prompt: "A distinguished older gentleman in a vintage study, surrounded by books and dim lighting, his face marked by wisdom and time. 8K, hyper-realistic, cinematic, post-production." The model with a lower guidance scale of $\kappa = 1.5$, which achieves a record-low one-step-generation FID of 8.15 and a competitive CLIP score of 0.304, produces images that are more diverse but align less closely with specific text details, such as "dim lighting." Conversely, the model with a higher guidance scale of $\kappa = 4.5$, achieving a high CLIP score of 0.322 and noted for state-of-the-art human preference scores (HPSv2) as shown in Table 2, presents a relatively high FID of 16.54, indicating less diversity but superior text alignment and visual quality.

Figure 8: Analogous to Figure 7, this plot compares two rows of images generated using two distinct guidance scales, both conditioning on the same text prompt: "poster art for the collection of the asian woman, in the style of gloomy, dark orange and white, dynamic anime, realistic watercolors, nintencore, weathercore, mysterious realism –ar 69:128 –s 750 –niji 5".

## A  BROADER IMPACT

The broader impact of our work is multifaceted. On one hand, it significantly reduces the energy required to operate state-of-the-art diffusion models, contributing to more sustainable AI practices. On the other hand, the ease of distilling and deploying models that might be trained on data with ques-

tionable content or intentions presents ethical challenges. Therefore, it is crucial for the community to engage in discussions on how to minimize risks while enhancing the benefits of such advancements. This involves developing robust guidelines and frameworks to govern the use and deployment of distilled models, ensuring they are used responsibly and ethically.

## B  RELATED WORK

Generative modeling of high-dimensional data has long been a focal point in machine learning research. This area primarily concentrates on replicating various data distributions: the original data distribution, conditional distributions influenced by labels, noisy or incomplete measurements, textual descriptions, or the joint distribution of data and other modalities. This has spurred the development of a diverse range of generative models and methodologies. Initially, these models were only capable of handling simpler, low-dimensional data such as $28 \times 28$ grayscale or binarized MNIST digits (Hinton et al., 2006; Salakhutdinov & Hinton, 2009; Vincent et al., 2010), vector-quantized local descriptors (Fei-Fei & Perona, 2005; Chong et al., 2009), or patches of natural and hyperspectral images (Zhou et al., 2009; Xing et al., 2012; Polatkan et al., 2015). Early models often utilized neural networks with stochastic binary hidden layers or shallow hierarchical Bayesian models, which are simpler to train but have limited capacities.

**Deep generative models.**   To tackle the generation of high-dimensional data, such as images comprising millions of pixels, substantial advancements in generative models have been made over the past decade. This period has marked the emergence of diverse deep generative models, including variational auto-encoders (VAEs) (Kingma & Welling, 2014; Rezende et al., 2014), normalizing flows (Papamakarios et al., 2019), generative adversarial networks (GANs) (Goodfellow et al., 2014; Reed et al., 2016; Karras et al., 2019; Wang et al., 2023b), autoregressive models (Gregor et al., 2015; Mansimov et al., 2015), and diffusion models (Sohl-Dickstein et al., 2015; Song & Ermon, 2019; Ho et al., 2020; Song & Ermon, 2020; Song et al., 2021b;a; Dhariwal & Nichol, 2021; Karras et al., 2022; Peebles & Xie, 2023; Zheng et al., 2024). Additionally, essential resources for creating effective T2I synthesis systems, such as large language models (Vaswani et al., 2017; Devlin et al., 2018; Radford et al., 2018; 2019; Raffel et al., 2020; He et al., 2020; Brown et al., 2020; Achiam et al., 2023; Touvron et al., 2023; Jiang et al., 2023), large vision-language models (Radford et al., 2021; Ilharco et al., 2021; Cherti et al., 2023), advanced visual tokenization and compression (Rolfe, 2016; van den Oord et al., 2017; Esser et al., 2021; Rombach et al., 2022; Podell et al., 2024), and extensive training datasets (Thomee et al., 2016; Changpinyo et al., 2021; Schuhmann et al., 2022), have transitioned from proprietary tools of major tech companies to publicly available assets.

Previously, T2I models primarily utilized GANs and focused on small-scale, object-centric domains like flowers, birds, and face images (Reed et al., 2016; Zhang et al., 2017; Xu et al., 2018; Zhang et al., 2020). Powered by text encoders that leverage pretrained large language models or vision-language models, which offer profound language comprehension and extract semantically rich latent representations, and supported by an extensive collection of text-image pairs, three main families of generative models—GANs (Zhou et al., 2022; Sauer et al., 2022; Kang et al., 2023), autoregressive models (Ramesh et al., 2021; Zhang et al., 2021; Ding et al., 2021; Gafni et al., 2022; Yu et al., 2022; Chang et al., 2023), and diffusion models (Nichol et al., 2022; Ramesh et al., 2022; Saharia et al., 2022; Rombach et al., 2022; Xu et al., 2022; Wang et al., 2023a; Qin et al., 2023)—have effectively capitalized on these technological advancements. These developments have facilitated the creation of T2I synthesis systems that demonstrate exceptional photorealism and sophisticated language understanding.

**Acceleration of diffusion-based generation.**  Early-stage diffusion models were notably slow in sampling, spurring extensive research aimed at speeding up the reverse diffusion process. Researchers have approached this by interpreting diffusion models through stochastic or ordinary differential equations and using advanced numerical solvers to enhance efficiency (Song et al., 2021c;a; Liu et al., 2022a; Lu et al., 2022b; Zhang & Chen, 2023; Karras et al., 2022). Additional strategies include truncating the diffusion chain to initiate generation from more structured distributions (Pandey et al., 2022; Zheng et al., 2023b; Lyu et al., 2022), integrating these models with GANs to boost generation speed (Xiao et al., 2022; Wang et al., 2023b), and exploring flow matching in diffusion modeling (Liu et al., 2022b; Lipman et al., 2022; Albergo et al., 2023). More recently, research has pivoted towards distilling reverse diffusion chains to refine and expedite the generation process, a direction

that continues to evolve with new methodologies and insights (Luhman & Luhman, 2021; Salimans & Ho, 2022; Zheng et al., 2023a; Meng et al., 2023; Song et al., 2023; Song & Dhariwal, 2023; Kim et al., 2023; Sauer et al., 2023b; Xu et al., 2023; Yin et al., 2023; Luo et al., 2023c; Zhou et al., 2024).

**T2I diffusion distillation.** Recent efforts aim to accelerate the sampling process from pre-trained diffusion teachers like SD (Rombach et al., 2022). Sauer et al. (2023b) focused on distilling diffusion models into generators capable of one or two-step operations through adversarial training. Xu et al. (2023) introduced UFOGen, utilizing a time-step-dependent discriminator for generator initialization. Luo et al. (2023a) applied consistency distillation (Song et al., 2023) to text-guided latent diffusion models (Ramesh et al., 2022) for efficient, high-fidelity T2I generation. Building on the idea of using a pre-trained 2D T2I diffusion model for text-to-3D synthesis (Poole et al., 2022; Wang et al., 2023c), SwiftBrush (Nguyen & Tran, 2024) showcases its effectiveness of distilling pre-trained stable diffusion models. Distribution Matching Distillation (DMD) by Yin et al. (2023) further enhances distillation quality by adding a regression loss term.

**How does SiD-LSG differ from previous methods?** SiD-LSG introduces several unique features that distinguish it from previous T2I diffusion distillation methods. Firstly, similar to SwiftBrush (Nguyen & Tran, 2024), SiD-LSG is a data-free distillation method, which eliminates the need for original training datasets or synthetic data generated with SDE/ODE solvers during the distillation process. However, unlike SwiftBrush, SiD-LSG necessitates gradient backpropagation through the score networks, a critical step akin to gradient backpropagation through the discriminator in methods employing adversarial losses. Additionally, SiD-LSG applies CFG to both the training and evaluation of the fake score network. This is a departure from previous methods, which typically apply CFG only during the evaluation of the pretrained score network. Finally, SiD-LSG aims to minimize a model-based explicit score-matching loss, a type of Fisher divergence. In contrast, previous methods often focus on minimizing losses based on KL divergence, consistency, regression, GAN-based adversarial tactics, or a combination thereof.

## C FURTHER DISCUSSION ON LONG, SHORT, AND LONG-SHORT GUIDANCE

Regarding the relationship between the choice of $\kappa_2 = \kappa_3$ and the categorization into long, short, and long-short guidance, we clarify that various combinations of $\kappa_{1,2,3,4}$ can define these strategies:

**Long Strategy:** This strategy involves enhancing the teacher's CFG more than the fake score network during inference or making them equal but both larger than one. It is typically represented by $\kappa_1 = \kappa_2 = \kappa_3 = 1, \kappa_4 > 1$, but also includes configurations where $\kappa_1 = \kappa_2 = \kappa_3 = \kappa_4 > 1$.

**Short Strategy:** This approach aims to reduce the fake-score-network's CFG during evaluation compared to training or maintain them equal but greater than one. This strategy is exemplified by $\kappa_1 = \kappa_4 = 1, 0 < \kappa_2 = \kappa_3 < 1$, and can also be part of $\kappa_1 = \kappa_2 = \kappa_3 = \kappa_4 > 1$.

**Long-Short Strategy:** The "simplest" LSG configuration $\kappa_1 > 1, \kappa_2 = \kappa_3 = \kappa_4 = 1$ can be interpreted as incorporating both long and short strategies since increasing $\kappa_1$ for training the fake score network effectively implies that during inference, the fake score network is guided by a weaker CFG than both the teacher and its own training setting.

SiD-LSG employs a default setting where $\kappa_1 = \kappa_2 = \kappa_3 = \kappa_4 = \kappa > 1$. This implies that while the teacher's training CFG is 1 and evaluation CFG is $\kappa > 1$, the fake score network's training and evaluation CFG are both $\kappa > 1$, making the teacher's CFG during inference stronger than that of the fake score network.

**FID-CLIP Compromise:** We evaluate various CFG strategies based on the best FID achieved with fewer than 2.56M fake images used to distill generators on Stable Diffusion 1.5, alongside their corresponding CLIP scores. The results, whose trajectories are depicted in Figures 3 and 4 and detailed in Table 3 and Figure 9, indicate that the proposed LSG strategy achieves the best balance between lowering FID and enhancing CLIP scores.

## D TRAINING AND EVALUATION DETAILS

The hyperparameters tailored for our study are outlined in Table 4. It's important to note that the time and memory costs reported in Table 4 do not account for those incurred during periodic evaluations

Table 3: Comparison of different CFG strategies in terms of the best FID achieved when iterating with fewer than 2.56M fake images to distill the generators on Stable Diffusion 1.5 and their corresponding CLIP scores.

| CFG Strategy | $\kappa_1$ | $\kappa_2$ | $\kappa_3$ | $\kappa_4$ | FID | CLIP |
|---|---|---|---|---|---|---|
| Long CFG | 1 | 1 | 1 | 2 | 10.01 | 0.297 |
| | 1 | 1 | 1 | 2.5 | 10.98 | 0.303 |
| | 1 | 1 | 1 | 3 | 11.75 | 0.303 |
| | 1 | 1 | 1 | 3.5 | 12.82 | 0.304 |
| | 1 | 1 | 1 | 7.5 | 15.78 | 0.308 |
| No CFG | 1 | 1 | 1 | 1 | 15.49 | 0.269 |
| Short CFG | 1 | 0.5 | 0.5 | 1 | 11.09 | 0.299 |
| | 1 | 0.125 | 0.125 | 1 | 15.54 | 0.305 |
| "Simplest" LSG | 2 | 1 | 1 | 1 | 11.76 | 0.306 |
| | 3 | 1 | 1 | 1 | 16.38 | 0.304 |
| LSG | 1.5 | 1.5 | 1.5 | 1.5 | 10.66 | 0.295 |
| | 2 | 2 | 2 | 2 | 10.44 | 0.304 |
| | 3 | 3 | 3 | 3 | 13.88 | 0.310 |
| | 4.5 | 4.5 | 4.5 | 4.5 | 16.69 | 0.310 |

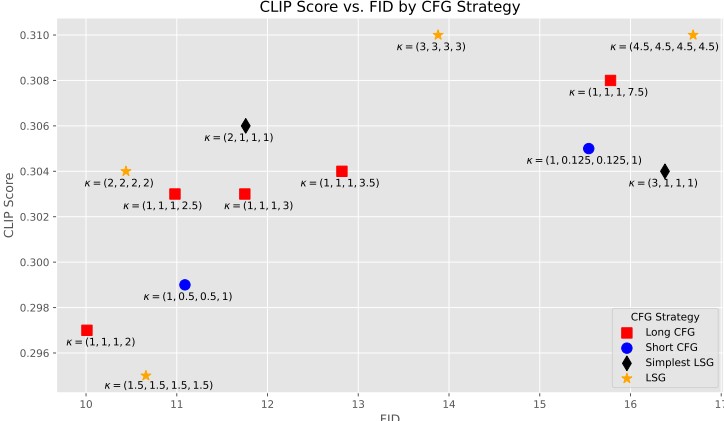

Figure 9: Comparison of different CFG strategies in terms of the best FID achieved when iterating with fewer than 2.56M images to distill the generators on Stable Diffusion 1.5 and their corresponding CLIP scores. The markers represent different CFG strategies: Long CFG is denoted by red squares, Short CFG by blue circles, the "simplest" LSG by black diamonds, and LSG by orange stars.

of the FID and CLIP scores of the single-step generator, nor do they include the resources used to save model checkpoints during the distillation process. These costs can vary significantly depending on the computing resources used, including the versions of CUDA and Flash Attention, as well as the storage platforms employed and the frequency of operations.

# E  LIMITATIONS AND COMPUTATIONAL REQUIREMENTS

**Memory and speed.** Table 4 in Appendix D offers a detailed examination of the computational resources required for the SiD distillation process employing various CFG strategies—Long, Short, "simplest" Long-Short, and the recommended Long-Short—across different NVIDIA GPU platforms (RTX-A5000 with 24GB, RTX-A6000 with 48GB, and H100 with 80GB). Key observations include:

SiD-LSG can be effectively operated on the RTX-A5000, which has 24GB of memory, by enabling xFormers (Lefaudeux et al., 2022) and switching to FP16 for model and gradient precision. On the RTX-A6000 with 48GB of memory, it runs using FP32 with xFormers enabled. However, on our available H100 with 80GB of memory, where xFormers were not supported at the time of our experiments, there was a noticeable increase in GPU memory consumption.

---

**Algorithm 1** SiD-LSG: Score identity Distillation with Long-Short classifier-free Guidance

---

**Input:** Pretrained score network $f_\phi$, generator $G_\theta$, fake score network $f_\psi$, $t_{\text{init}} = 625$, $t_{\min} = 20$, $t_{\max} = 979$, guidance scales $\kappa_1 = \kappa_2 = \kappa_3 = \kappa_4 = 1.5$

**Initialization** $\theta \leftarrow \phi, \psi \leftarrow \phi$

**repeat**

    Sample $\boldsymbol{z} \sim \mathcal{N}(0, \mathbf{I})$ and $\boldsymbol{c}$, replacing $\boldsymbol{c}$ with the embedding of an empty text string 10% of the time; Define $\boldsymbol{x}_g = G_\theta(\boldsymbol{z}, \boldsymbol{c}) = f_\theta(\sigma_{t_{\text{init}}} \boldsymbol{z}, t_{\text{init}}, \boldsymbol{c})$; Sample $t \in \{t_{min}, \ldots, t_{\max}\}$ and $\boldsymbol{\epsilon}_t \sim \mathcal{N}(0, \mathbf{I})$, and let $\boldsymbol{x}_t = a_t \boldsymbol{x}_g + \sigma_t \boldsymbol{\epsilon}_t$

    Update $\psi$ with $\psi = \psi - \eta \nabla_\psi \hat{\mathcal{L}}_\psi$, where

$$\hat{\mathcal{L}}_\psi = \frac{a_t^2}{\sigma_t^2} \|f_{\psi, \kappa_1}(\boldsymbol{x}_t, t, \boldsymbol{c}) - \boldsymbol{x}_g\|_2^2 = \|\boldsymbol{\epsilon}_{\psi, \kappa_1}(\boldsymbol{x}_t, t, \boldsymbol{c}) - \boldsymbol{\epsilon}_t\|_2^2$$

    Sample $\boldsymbol{z} \sim \mathcal{N}(0, \mathbf{I})$ and $\boldsymbol{c}$, and let $\boldsymbol{x}_g = G_\theta(\boldsymbol{z}, \boldsymbol{c}) = f_\theta(\sigma_{t_{\text{init}}} \boldsymbol{z}, t_{\text{init}}, \boldsymbol{c})$; Sample $t \in \{t_{min}, \ldots, t_{\max}\}$ and $\boldsymbol{\epsilon}_t \sim \mathcal{N}(0, \mathbf{I})$, compute $\omega_t$ with (9), and let $\boldsymbol{x}_t = a_t \boldsymbol{x}_g + \sigma_t \boldsymbol{\epsilon}_t$

    Update $G_\theta$ with $\theta = \theta - \eta \nabla_\theta \tilde{\mathcal{L}}_\theta$, where

$$\tilde{\mathcal{L}}_\theta = \frac{\omega(t) a_t^2}{\sigma_t^4} (f_{\phi, \kappa_4}(\boldsymbol{x}_t, t, \boldsymbol{c}) - f_{\psi, \kappa_2}(\boldsymbol{x}_t, t, \boldsymbol{c}))^T (f_{\psi, \kappa_3}(\boldsymbol{x}_t, t, \boldsymbol{c}) - \boldsymbol{x}_g)$$

$$= \frac{\omega(t)}{\sigma_t^2} (\boldsymbol{\epsilon}_{\psi, \kappa_2}(\boldsymbol{x}_t, t, \boldsymbol{c}) - \boldsymbol{\epsilon}_{\phi, \kappa_4}(\boldsymbol{x}_t, t, \boldsymbol{c}))^T (\boldsymbol{\epsilon}_t - \boldsymbol{\epsilon}_{\psi, \kappa_3}(\boldsymbol{x}_t, t, \boldsymbol{c}))$$

**until** processing 10M fake images or the training budget is exhausted

**Output:** $G_\theta$

---

The recommended LSG strategy, which provides an improved balance between FID and CLIP scores, demands approximately 20% more computation time per iteration and exhibits about 10% more peak memory usage on the H100 compared to the Long or Short strategies. This underlines a trade-off between achieving performance enhancements and managing resource utilization.

**FP16 versus FP32.** SiD-LSG can be trained under FP16 mixed precision, which significantly conserves memory and enhances processing speed, as detailed in Table 4. Although this reduced precision in optimization leads to quicker improvements in both FID and CLIP scores, it also restricts the potential for achieving the lowest FID and highest CLIP scores compared to results under FP32. These effects are demonstrated in the ablation study shown in Figure 10. Additionally, FP16 operation is less stable and requires the use of gradient clipping to maintain training stability. For instance, we employ `torch.nn.utils.clip_grad_value_(G.parameters(), 1)` to prevent sudden model divergence—a precaution that is not necessary with FP32.

Further investigation is needed to optimize FP16 performance to match that of FP32. This may involve refining loss scaling techniques, updating packages like Flash Attention or XFormers, or implementing an effective warmup period with FP16 before transitioning to FP32. We plan to address these challenges in our future studies.

We note that our initial experimental platform did not provide proper support for FlashAttention (Dao et al., 2022; Dao, 2023), which adversely affected our FP16 results. Now that we have established proper support for FlashAttention, we are keen to further explore the potential of FP16 in distilling SiD-LSG, especially to determine if it can match FP32's performance at a lower cost. We have updated Table 4 displayed above to reflect significant memory reductions and a noticeable acceleration in iteration speed under FP16, facilitated by the availability of FlashAttention. This enhancement would enable us to use larger batch sizes per GPU under FP16, further improving time efficiency.

**Reaching a performance plateau.** Zhou et al. (2024) demonstrate through extensive comparisons that the SiD distilled one-step generator can reduce the FID at an exponential decay rate, showing a log-log linear relationship between the number of iterations and FID. This approach can match or even surpass the performance of both unconditional and label-conditional teacher diffusion models trained under the EDM framework (Karras et al., 2022), provided sufficient training.

However, results shown in Figures 4 and 5 indicate that the SiD-LSG distilled one-step generator on SD1.5 quickly reaches a performance plateau in reducing FID and/or increasing CLIP scores, particularly when the LSG scale exceeds 2. Additionally, data from Table 1 demonstrate that after processing 10M images (approximately 20k iterations at a batch size of 512), SiD-LSG still does not match the text-image alignment performance of the teacher model, which achieves higher CLIP scores after 50 generation steps. Notably, by reducing the LSG scale to 1.5 and doubling the training duration, SiD-LSG achieves a record-low data-free FID of 8.15, establishing a new benchmark among all data-free diffusion distillation methods. This achievement also surpasses the teacher model's FID

Table 4: Hyperparameter settings and comparison of distillation time and memory usage between different long and short guidance strategies of SiD. Note in order to run the long-short guidance (LSG) with $\kappa_1 = \kappa_2 = \kappa_3 = \kappa_4 > 1$, we need to turn off the EMA network for some cases (indicated with "no EMA"), otherwise it will be out of memory.

| Computing platform | Hyperparameters | Long Strategy | Short Strategy | Long-Short | Long-Short |
|---|---|---|---|---|---|
| General Settings | CFG scales | $\kappa_4 > 1$ $\kappa_1 = \kappa_2 = \kappa_3 = 1$ | $0 < \kappa_2 = \kappa_3 < 1$ $\kappa_1 = \kappa_4 = 1$ | $\kappa_1 > 1$ $\kappa_2 = \kappa_3 = \kappa_4 = 1$ | $\kappa_{1,2,3,4} > 1$ $\kappa_1 = \kappa_2 = \kappa_3 = \kappa_4$ |
| | Batch size | | | 512 | |
| | Learning rate | | | 1e-6 | |
| | Half-life of EMA | | | 50k images | |
| | Optimizer under FP32 | | Adam ($\beta_1 = 0$, $\beta_2 = 0.999$, $\epsilon$ = 1e-8) | | |
| | Optimizer under FP16 | | Adam ($\beta_1 = 0$, $\beta_2 = 0.999$, $\epsilon$ = 1e-6) | | |
| | $\alpha$ | | | 1.0 | |
| | Time parameters | | $(t_{\min}, t_{\text{init}}, t_{\max}) = (20, 625, 979)$ | | |
| RTX-A5000 (24G), FP16 | xFormers available and enabled | | | Yes | |
| | Batch size per GPU | | | 1 | |
| | # of GPUs | | | 8 | |
| | # of gradient accumulation round | | | 64 | |
| | Max memory in GB allocated | 22.9 | 22.9 | 22.6 | 22.0 (no EMA) |
| | Max memory in GB reserved | 23.0 | 23.0 | 23.0 | 22.1 (no EMA) |
| | Time in seconds per 1k images | 74 | 75 | 80 | 102 |
| | Time in hours per 1M images | 21 | 21 | 22 | 29 |
| RTX-A6000 (48G), FP32 | xFormers available and enabled | | | Yes | |
| | Batch size per GPU | | | 1 | |
| | # of GPUs | | | 8 | |
| | # of gradient accumulation round | | | 64 | |
| | Max memory in GB allocated | 45.7 | 45.7 | 45.0 | 45.8 (no EMA) |
| | Max memory in GB reserved | 45.7 | 45.7 | 45.9 | 46.0 (no EMA) |
| | Time in seconds per 1k images | 365 | 365 | 366 | 502 |
| | Time in hours per 1M images | 102 | 102 | 102 | 139 |
| H100 (80G), FP16 | xFormers available and enabled | | | No | |
| | Batch size per GPU | | | 4 | |
| | # of GPUs | | | 8 | |
| | # of gradient accumulation round | | | 16 | |
| | Max memory in GB allocated | 55.8 | 55.8 | 47.7 | 63.9 |
| | Max memory in GB reserved | 57.3 | 58.3 | 49.2 | 65.4 |
| | Time in seconds per 1k images | 17 | 17 | 18 | 23 |
| | Time in hours per 1M images | 5 | 5 | 5 | 6 |
| H100 (80G), FP16 | Flash Attention available and enabled | | | Yes | |
| | Batch size per GPU | | | 4 | |
| | # of GPUs | | | 8 | |
| | # of gradient accumulation round | | | 16 | |
| | Max memory in GB allocated | 32.3 | 32.2 | 29.2 | 35.2 |
| | Max memory in GB reserved | 32.6 | 32.4 | 29.6 | 35.4 |
| | Time in seconds per 1k images | 12 | 12 | 12 | 15 |
| | Time in hours per 1M images | 3 | 3 | 3 | 4 |
| H100 (80G), FP32 | xFormers available and enabled | | | No | |
| | Batch size per GPU | | | 1 | |
| | # of GPUs | | | 8 | |
| | # of gradient accumulation round | | | 64 | |
| | Max memory in GB allocated | 58.9 | 57.4 | 53.4 | 62.9 |
| | Max memory in GB reserved | 60.0 | 59.0 | 54.0 | 64.0 |
| | Time in seconds per 1k images | 74 | 74 | 76 | 90 |
| | Time in hours per 1M images | 21 | 21 | 21 | 25 |

of 8.78, which was obtained with a CFG scale of 3 and 50 generation steps. However, this reduction in the LSG scale to 1.5 also results in a significant decline in its CLIP score.

This suggests significant potential for further performance enhancements, possibly by extending beyond single-step generation, increasing the model size, or incorporating real data and additional regression or adversarial losses into the distillation process. These avenues for improvement will be one of the focuses of future studies.

**Additional ablation study of the guidance strategies.** At the beginning of our research, we evaluated various combinations of $\kappa_{1,2,3,4}$, quickly discarding those that converged too slowly or diverged. Specifically, the setting $\kappa_1 = 1$, $\kappa_2 = \kappa_3 > 1$, and $\kappa_4 \in \{\kappa_2, 1\}$ was eliminated early due to its suboptimal performance, as depicted in Figure 11.

## F  PROMPTS AND ADDITIONAL EXAMPLE IMAGES

We list the prompts used in Figure 1 as follows:

1. A distinguished older gentleman in a vintage study, surrounded by books and dim lighting, his face marked by wisdom and time. 8K, hyper-realistic, cinematic, post-production.

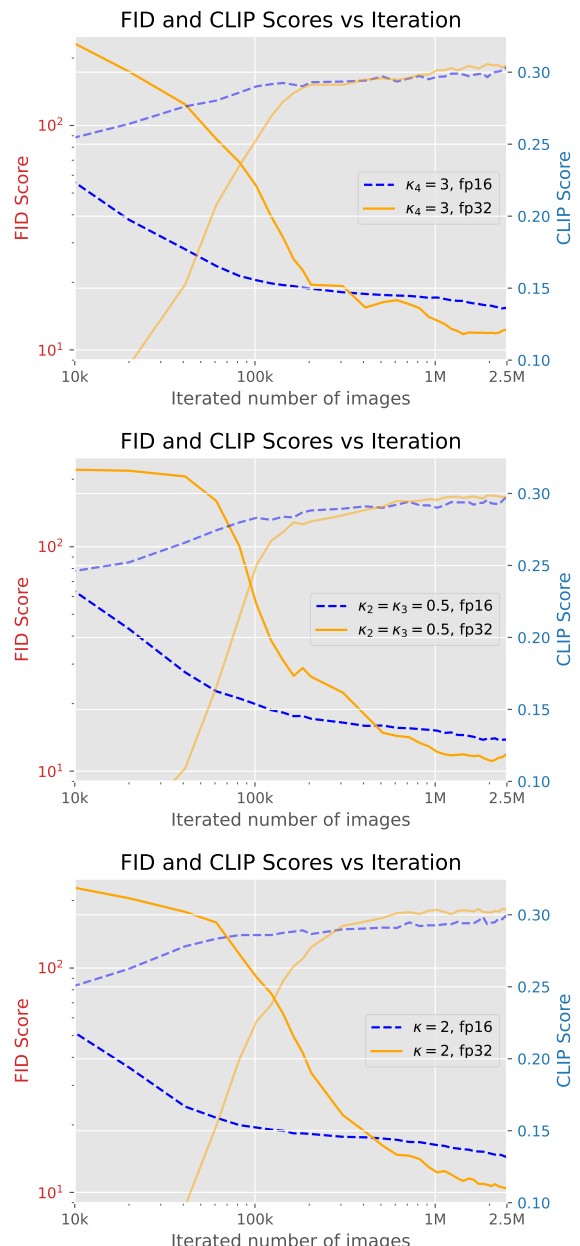

Figure 10: Comparison between FP16 and FP32 under three different guidance strategies. Top: Long strategy with $\kappa_1 = \kappa_2 = \kappa_3 = 1$ and $\kappa_4 = 3$. Middle: Short strategy with $\kappa_1 = \kappa_4 = 1$ and $\kappa_2 = \kappa_3 = 0.5$. Bottom: Long-short guidance (LSG) with $\kappa_1 = \kappa_2 = \kappa_3 = \kappa_4 = 2$.

2. saharian landscape at sunset , 4k ultra realism, BY Anton Gorlin, trending on artstation, sharp focus, studio photo, intricate details, highly detailed, by greg rutkowski.

3. chinese red blouse, in the style of dreamy and romantic compositions, floral explosions –ar 24:37 –stylize 750 –v 6

4. Digital 2D, Miyazaki's style, ultimate detailed, tiny finnest details, futuristic, sci-fi, magical dreamy landscape scenery, small cute girl living alone with plushified friendly big tanuki in the gigantism of wilderness, intricate round futuristic simple multilayered architecture, habitation cabin in the trees, dramatic soft lightning, rule of thirds, cinematic.

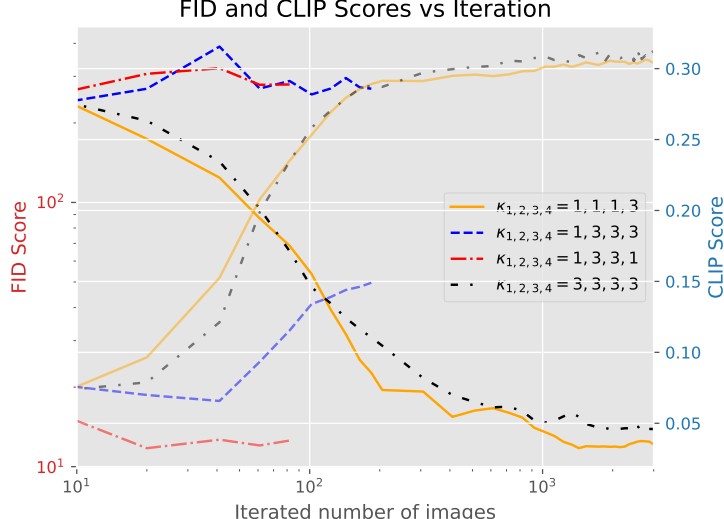

Figure 11: This figure shows the progression of FID and CLIP scores during an ablation study of distilling SD1.5 using SiD-LSG, featuring four different sets of $\kappa_{1,2,3,4}$ values. Settings that converged too slowly were terminated early.

5. poster art for the collection of the asian woman, in the style of gloomy, dark orange and white, dynamic anime, realistic watercolors, nintencore, weathercore, mysterious realism –ar 69:128 –s 750 –niji 5

6. A fantasy-themed portrait of a female elf with golden hair and violet eyes, her attire shimmering with iridescent colors, set in an enchanted forest. 8K, best quality, fine details.

7. 'very beautiful girl in bright leggings, white short top, charismatic personality, professional photo, style of jessica drossin, super realistic photo, hyper detail, great attention to skin and eyes, professional photo.

8. (steampunk atmosphere, a stunning girl with a mecha musume aesthetic, adorned in intricate cyber gogle,) digital art, fractal, 32k UHD high resolution, highres, professional photography, intricate details, masterpiece, perfect anatomy, cinematic angle , cinematic lighting, (dynamic warrior pose:1)

9. (Pirate ship sailing into a bioluminescence sea with a galaxy in the sky), epic, 4k, ultra.

10. tshirt design, colourful, no background, yoda with sun glasses, dancing at a festival, ink splash, 8k.

We list the prompts used in Figure 2, which are taken from the COCO-2014 validation set, as follows:

1. many cars stuck in traffic on a high way

2. an old blue car with a surfboard on top', 3. 'a sole person sits in the front pew of a large church.

3. a shot of the hollywood sign at santa monica blvd.

4. a bunch of flowers, in front of a forest.

5. there is some sort of vegetables in a bowl

6. a man in a pink shirt stands staring against a green wall.

7. small girl in green shirt holding a slice of pizza to her face

8. two dogs sitting in the back seat of a car looking out the windwo

We list the prompts used in Figure 6 as follows:

1. Half-length head portrait of the goddess of autumn with wheat ears on her head, depicted as dreamy and beautiful, by wlop.

2. Walter White dressed as a medieval-style king.

3. A serene meadow with a tree, river, bridge, and mountains in the background under a slightly overcast sunrise sky.

4. A closeup portrait of a gray owl with spreaded wings attacking in cinematic lighting, digital painting by Greg Rutkowski used as album cover art on Artstation.

5. A hyena fursona sitting in the grass in a savannah at sunset.

6. A puppy staring through a red sectioned window.

We list the prompts used in Figure 12 as follows:

1. A fantasy-themed portrait of a female elf with golden hair and violet eyes, her attire shimmering with iridescent colors, set in an enchanted forest. 8K, best quality, fine details.

2. pumpkins, autumn sunset in the old village, cobblestone houses, streets, plants, flowers, entrance, realistic, stunningly beautiful

3. "Highly detailed mysterious egyptian (sphynx cat), skindentation:1.2, bright eyes, ancient egypt pyramid background, photorealistic, (hyper-realistic:1.2), cinematic, masterpiece:1.1, cinematic lighting"

4. "vw bus, canvas art, abstract art printing, in the style of brian mashburn, light red and light brown, theo prins, charming character illustrations, pierre pellegrini, vintage cut-and-paste, rusty debris –ar 73:92 –stylize 750 –v 6"

5. painterly style, seductive female League of legends Jinx character fighting at war, raging, crazy smile, crazy eyes, rocket lancher, guns, crazy face expression, character design, body is adorned with glowing golden runes, intense green aura around her, body dynamic epic action pose, intricate, highly detailed, epic and dynamic composition, dynamic angle, intricate details, multicolor explosion, blur effect, sharp focus, uhd, hdr, colorful shot, stormy weather, tons of flying debris around her, dark city background, modifier=CarnageStyle, color=blood_red, intensity=1.6

6. A charismatic chef in a bustling kitchen, his apron dusted with flour, smiling as he presents a beautifully prepared dish. 8K, hyper-realistic, cinematic, post-production.

7. A young adventurer with tousled hair and bright eyes, wearing a leather jacket and a backpack, ready to explore distant lands. 8K, hyper-realistic, cinematic, post-production.

8. "A watercolor painting of a vibrant flower field in spring, with a rainbow of blossoms under a bright blue sky. 8K, best quality, fine details.",

9. "digital art of a beautiful tiger pokemon under an apple tree, cartoon style,Matte Painting,Magic Realism,Bright colors,hyper quality,high detail,high resolution, –video –s 750 –v 6.0 –ar 1:2"

10. "painterly style, Goku fighting at war, raging, blue hair, character design, body is adorned with glowing golden runes, yellow aura around him, body dynamic epic action pose, intricate, highly detailed, epic and dynamic composition, dynamic angle, intricate details, multicolor explosion, blur effect, sharp focus, uhd, hdr, colorful shot, stormy weather, tons of flying debris around him, dark city background, modifier=CarnageStyle, color=blood_red, intensity=1.6"

11. A stunning steampunk city with towering skyscrapers and intricate clockwork mechanisms, gears and pistons move in a complex symphony, steam billows from chimneys, airships navigate the bustling skylanes, a vibrant metropolis

12. "Samurai looks at the enemy, stands after the battle, fear and horror on his face, tired and beaten, sand on his face mixed with sweat, an atmosphere of darkness and horror, hyper realistic photo, In post - production, enhance the details, sharpness, and contrast to achieve the hyper - realistic effect"

13. A portrait of an elemental entity with strong rim lighting and intricate details, painted digitally by Alvaro Castagnet, Peter Mohrbacher, and Dan Mumford

14. "A regal female portrait with an ornate headdress decorated with colorful gemstones and feathers, her robes rich with intricate designs and bright hues. 8K, best quality, fine details.",

15. "A detailed painting of Atlantis by multiple artists, featuring intricate detailing and vibrant colors.",

16. "A landscape featuring mountains, a valley, sunset light, wildlife and a gorilla, reminiscent of Bob Ross's artwork.

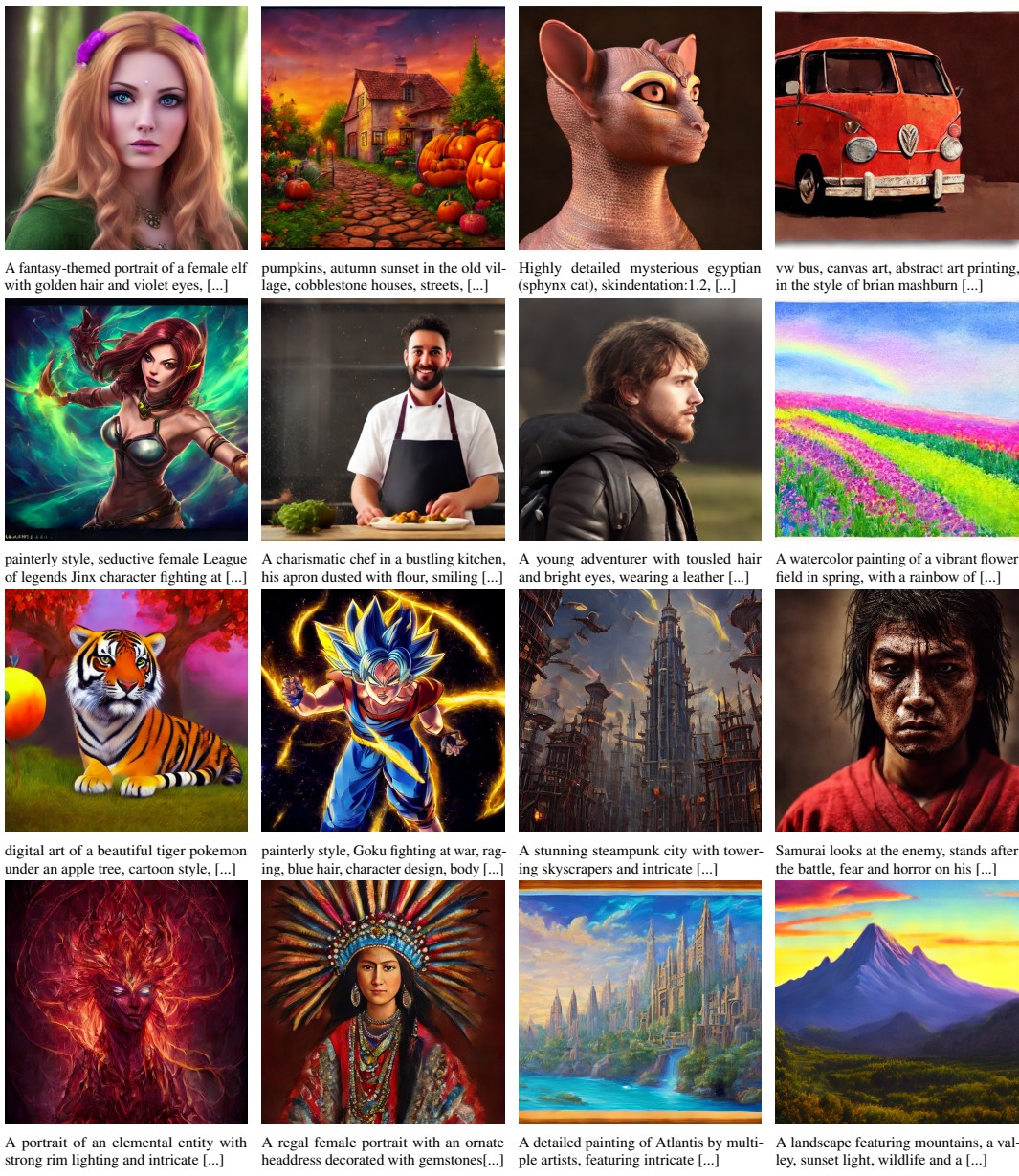

Figure 12: More examples from the one-step generator distilled from Stable Diffusion 2.1-base using the proposed method: Score identity Distillation with Long-Short Guidance.

