# OpenReview forum: "Guided Score identity Distillation for Data-Free One-Step Text-to-Image Generation"
_ICLR.cc/2025/Conference — ICLR 2025 Poster_

### Official Review · Reviewer_fY7L · 2024-11-03

**Soundness:** 4
**Presentation:** 3
**Contribution:** 3
**Rating:** 6
**Confidence:** 2

**Summary:**

This paper mainly studies how to accelerate the diffusion-based generative model to generate high-quality images. The paper presents a novel method called Guided Score Identity Distillation with Long-Short Guidance (SiD-LSG) for data-free one-step text-to-image generation, which enhances the Score Identity Distillation (SiD) technique by incorporating Long and Short Classifier-Free Guidance (LSG).

**Strengths:**

1.  The main contribution is The proposed method enhances the Score Identity Distillation technique by incorporating Classifier-Free Guidance , although these two techniques have been proposed in other studies.

2. By reducing the number of steps required for image generation to one, the method significantly improves the efficiency of the generation process, and the proposed method achieves state-of-the-art performance in terms of Fréchet Inception Distance (FID) on the COCO-2014 validation set, setting a new benchmark for data-free one-step distillation.

**Weaknesses:**

1.  Regarding training efficiency, SiD needs to generate false data for training, and adding CFG will further increase the model overhead, resulting in more training time and resources.

2. The data-free nature of the SiD and SID-LSG means it does not leverage real-world data to further refine the generated images, which could potentially improve photorealism and text alignment.

**Questions:**

If a model itself does not use CFG, then the proposed method will not work？ For example, FLUX does not use CFG for inference but uses word embedding instead. Just open discussion.

---

> ### Author Response · Authors · 2024-11-17
> **Clarifications on CFG, training efficiency, and FLUX**
>
> We thank the reviewer for recognizing the state-of-the-art performance of the proposed SiD-LSG.
>
> **Novelty of LSG:** We would like to emphasize that while CFG has been utilized previously, its application to the training of the fake score network represents a novel aspect of our Long-Short Guidance (LSG) strategy. This combination of SiD and LSG has proven effective in achieving a better compromise between low FID and high CLIP scores, as further illustrated in the newly added Table 3 and Figure 9 in the Appendix.
>
> **Training Efficiency:** It is essential to clarify that generating fake images and incorporating CFG for distillation are standard practices among existing distillation methods. As depicted in Figures 2-5, SiD-LSG swiftly generates photo-realistic images and continues to lower FID and enhance CLIP scores at an approximately exponential rate before signs of plateauing appear. A key advantage of SiD-LSG is its seemingly consistent improvement as computational resources increase—there is no observed risk of FID (CLIP) scores worsening after certain iterations. However, due to the exponential rate of improvement (approximately log-log linear in FID (CLIP) vs. iterations) observed in our experiments, the rate of progress naturally diminishes over time in linear terms.
>
> **Role of CFG:** As illustrated in Figure 3 (Right) and the newly added Table 3 in the Appendix, our model functions without CFG but yields unsatisfactory CLIP scores, indicating suboptimal alignment of the generated images with the provided text prompts. This supports prior observations on the crucial role of CFG in improving text-image alignment in established text-image diffusion models, such as SD1.5 and SD2.1-base. Given that SiD-LSG distills from these models, its dependence on CFG is anticipated.
>
> **CFG in FLUX:** Regarding the question about FLUX, it is essential to note that FLUX is not yet fully open-access, and no detailed technical report is currently available that elaborates on its design, training, and inference details. Our response is based on publicly available information.
>
> Based on available resources, FLUX still relies on CFG to enhance its text-image alignments:
> - FLUX.1 [pro] allows a guidance scale between 2 and 5, which "Controls the balance between adherence to the text prompt and image quality/diversity. Higher values make the output more closely match the prompt but may reduce overall image quality. Lower values allow for more creative freedom but might produce results less relevant to the prompt," according to https://replicate.com/black-forest-labs/flux-pro.
> - FLUX.1 [dev] allows a guidance scale between 0 and 10, as stated on https://replicate.com/black-forest-labs/flux-dev. FLUX.1 [dev], described as an open-weight, guidance-distilled model for non-commercial applications on https://blackforestlabs.ai/, is directly distilled from FLUX.1 [pro] and offers similar quality and prompt adherence capabilities while being more efficient than a standard model of the same size.
> - FLUX.1 [schnell] is noted as a guidance and step-distilled variant on https://github.com/black-forest-labs/flux.
>
> It would be intriguing to explore whether SiD-LSG could be applied to distill FLUX [pro], [dev], and [schnell]. Theoretically, SiD-LSG is well-suited for this task as it does not require access to the training data used for FLUX. Practically, we must consider the licensing implications of FLUX and address technical challenges such as adapting from the Distributed Data Parallel (DDP) framework, which SiD-LSG currently uses, to alternative distributed training frameworks that allow for model splitting across GPUs. We are keen to pursue this exploration in our future work, contingent upon a clear understanding of licensing implications and the availability of computing resources.

---

> > ### Comment · Reviewer_fY7L · 2024-11-18
> > **Reply to the author**
> >
> > Thanks for the author's response, which solved many of my concerns. I decided to improve the score.

---

> > > ### Author Response · Authors · 2024-11-18
> > >
> > > We thank the reviewer for reading our response and adjusting the score. Please do not hesitate to reach out to us should any new questions arise.

---

### Official Review · Reviewer_F9u6 · 2024-11-03

**Soundness:** 4
**Presentation:** 4
**Contribution:** 3
**Rating:** 8
**Confidence:** 4

**Summary:**

The authors propose an enhancement to Score Identity Distillation (SiD), a recently introduced method for distilling a pre-trained diffusion network into a single-step generator. This paper’s method specifically targets pre-trained text-to-image models based on DDPM, adapting SiD to distill two epsilon-prediction networks: SD1.5 and SD2.1.base.

Following the SiD framework, the one-step generator ($\theta$) is optimized by training an additional fake multi-step generator ($\psi$). Furthermore, classifier-free guidance (CFG) is integrated into the distillation process, allowing for varied CFG values that emphasize different target objectives. This added flexibility contributes to enhanced model performance.

The results achieve state-of-the-art FID scores and demonstrate strong CLIP scores on the COCO-2014 validation prompts.

**Strengths:**

- The paper is well written and easy to follow
- The novelty and relation to SiD is clear.
- The theory and mathematical background is solid.
- The method is novel and important.
- The ablation is comprehensive.
- The results of the methods are SoTA.

**Weaknesses:**

- Given SiD the method novelty is limited to the adaptation of the SiD theory to DDPM and t2i models.
- The distillation is on relatively old pre-trained networks.

**Questions:**

- Does this approach valid for v-prediction networks? how does this alter the theory?

---

> ### Author Response · Authors · 2024-11-17
> **Response to Reviewer F9u6**
>
> We appreciate the reviewer for providing a clear list of the contributions of this paper and thank you for your support.
>
> We would like to highlight a novel aspect of this work: the introduction of long and short CFG strategies, which deviate from the traditional long strategy typically employed in this field. To illustrate this, Table 3 and Figure 9 newly added into the Appendix demonstrate that the proposed Long-Short Guidance (LSG) achieves a superior balance in maximizing CLIP scores and minimizing FID, as evidenced by a larger area under the CLIP-FID curve compared to other strategies.
>
> Our focus was on the well-benchmarked SD1.5 and SD2.1-base, and we hope that our open-source code and model checkpoints will enable others to extend our method to both existing and future diffusion/flow-based generative models. We plan to further enhance SiD-LSG, with anticipated demonstrations of these improvements not only on SD1.5 and SD2.1-base but also on more advanced models such as Stable Diffusion XL/3.5 and other open-source diffusion/flow-matching models based on Unet/DiT architectures.
>
> The current approach is also applicable to v-prediction. Using the Diffuser library to transition from epsilon prediction to v-prediction, the only two modifications involve changing the noise schedule from DDPM to DDIM and adjusting the loss for the fake score network as follows:
>
> ```python
> #Sudo code:
> images = G_theta(z, contexts).detach()
> noise_fake = sid_sd_denoise(unet=fake_score_ddp, images=images, noise=noise, contexts=contexts, timesteps=timesteps,
>                             noise_scheduler=noise_scheduler,
>                             text_encoder=text_encoder, tokenizer=tokenizer,
>                             resolution=resolution, dtype=dtype, predict_x0=False, guidance_scale=cfg_train_fake)
> if noise_scheduler.config.prediction_type == "v_prediction":
>     target = noise_scheduler.get_velocity(images, noise, timesteps)
>     loss = (noise_fake - target) ** 2
>     snr = compute_snr(noise_scheduler, timesteps)
>     loss = loss * snr / (snr + 1)
> else:
>     loss = (noise_fake - noise) ** 2
>
> loss = loss.sum().mul(loss_scaling / batch_gpu_total)
> ```
>
> Under this adaptation, the SiD-LSG algorithm can then be directly applied to distill Stable-Diffusion-2-1 that uses v-prediction.

---

> > ### Comment · Reviewer_F9u6 · 2024-11-26
> > **Reply to the author**
> >
> > I thank the authors for the detailed reply.
> > After reading other reviews and the author response, my score remains the same and I recommend on accepting the paper.

---

### Official Review · Reviewer_8grk · 2024-11-04

**Soundness:** 3
**Presentation:** 2
**Contribution:** 3
**Rating:** 6
**Confidence:** 2

**Summary:**

The paper proposes a new distillation technique (SiD-LSG) for data-free one-step text-to-image generation. First, it adapts Score identity Distillation (SiD) to work with text-to-image models. Second, it proposes a Long and Short Classifier-Free Guidance (LSG) scheme to improve the distillation effectiveness. SiD-LSG shows state-of-the-art distillation results in a data-free setting. With a small CFG, it achieves impressive FID scores (8.15 for SD1.5 and 9.52 for SD2.1). With a large CFG, it achieves high CLIP scores, with sacrification in FID ones.

**Strengths:**

- The paper adapts Score identity Distillation (SiD) to work with text-to-image models.
- SiD-LSG shows state-of-the-art distillation results in a data-free setting. With a small CFG, it achieves impressive FID scores (8.15 for SD1.5 and 9.52 for SD2.1). With a large CFG, it achieves high CLIP scores, with sacrification in FID ones.
- The paper presents a comprehensive set of experiments.

**Weaknesses:**

- L268-269: The claim "we are the first to incorporate CFG into the training of the fake score network" is incorrect. SwiftBrush uses CFG 4.5 for both teachers (see Implementation Details in Section 4.1 in their paper).
- While the paper proposes an interesting short guidance setting ($\kappa 2 = \kappa 3 < 1$), it does not apply to the long and short guidance ($\kappa 2 = \kappa 3 > 1$). I do not see the connection between the short guidance setting and the long and short guidance setting.
- In the end, the long and short guidance scheme is very simple and uninteresting ($\kappa 1 = \kappa 2 = \kappa 3 = \kappa 4 = 1.5). It just applies the same CFG for both teachers, which SwiftBrush already did.
- Some recent works (SwiftBrush v2, DMD2) should be mentioned and compared with.
- The training protocol is expensive. The batch size is large (512 by default), requiring 64 gradient accumulation rounds on H100 (80G), FP32 setting. SiD-LSG necessitates gradient backpropagation through the score networks, which requires massive memory usage. Hence, it can only use the batch size per GPU as 1 on H100 (80G), FP32 setting.
- Fig 3, 4, 6, 9: The curves for FID scores and CLIP scores are hard to differentiate. They have the same color and with just a small transparency difference. The authors should improve these figures.
- Fig. 6, which is discussed in Section 3.2, is in the Appendix. It is not good. The authors should move the figure to the main paper and squeeze the text to fit in the 10-page limit.

**Questions:**

See the weaknesses.

---

> ### Author Response · Authors · 2024-11-13
>
> Thank you for your constructive feedback. Before addressing each of your concerns in detail, we’d like to clarify a key misunderstanding regarding SwiftBrush, which we had initially found unclear ourselves. Specifically, we sought clarification from the authors on their statement: *"During training, a guidance scale of 4.5 is used for both teachers."*
>
> The authors confirmed that *“Classifier-Free Guidance (CFG) was not applied to the LoRA teacher in Equation 6. We directly use the output from the LoRA teacher and subtract it with random Gaussian noise ε.”*
>
> In other words, CFG in SwifthBrush is used during the inference phase of the fake score network but is not applied during training (the "training" in SwiftBrush refers to the training of the generator). Thus, we stand by our claim that **we are the first to incorporate CFG into the training of the fake score network.**

---

> > ### Author Response · Authors · 2024-11-18
> > **Further clarifications**
> >
> > We appreciate the reviewer's recognition of the state-of-the-art performance achieved by SiD-LSG and acknowledge the need for further clarification. We have incorporated the clarifications provided below into the revision.
> >
> > Regarding the relationship between the choice of $\kappa_2=\kappa_3$ and the categorization into long, short, and long-short guidance, we clarify that various combinations of $\kappa_{1,2,3,4}$ can define these strategies:
> >
> > - **Long Strategy**: This strategy involves enhancing the teacher’s CFG more than the fake score network during inference or making them equal but both larger than one. It is typically represented by $\kappa_1=\kappa_2=\kappa_3=1, \kappa_4>1$, but also includes configurations where $\kappa_1=\kappa_2=\kappa_3=\kappa_4>1$.
> >
> > - **Short Strategy**: This approach aims to reduce the fake-score-network’s CFG during evaluation compared to training or maintain them equal but greater than one. This strategy is exemplified by $\kappa_1=\kappa_4=1, 0<\kappa_2=\kappa_3<1$, and can also be part of $\kappa_1=\kappa_2=\kappa_3=\kappa_4>1$.
> >
> > Additionally, the configuration $\kappa_1>1, \kappa_2=\kappa_3=\kappa_4=1$ can be interpreted as incorporating both long and short strategies since increasing $\kappa_1$ for training the fake score network effectively implies that during inference, the fake score network is guided by a weaker CFG than both the teacher and its own training setting.
> >
> > Unlike the SwiftBrush approach where $\kappa_1=1$, SiD-LSG employs a setting where $\kappa_1= \kappa_2=\kappa_3=\kappa_4=\kappa>1$. This implies that while the teacher’s training CFG is 1 and evaluation CFG is $\kappa>1$, the fake score network's training and evaluation CFG are both $\kappa>1$, making the teacher’s CFG during inference stronger than that of the fake score network.
> >
> > In our analysis, we referenced SwiftBrush and DMD. Given their recent updates and non-data-free nature, we consider SwiftBrushv2 and DMD2 as concurrent works and therefore deemed it unnecessary to include comparisons with them.
> >
> > We have repositioned Figure 6 back to the main paper. We have maintained consistent color coding for FID and CLIP curves under the same settings and added clarifications to the figure caption to prevent any confusion.
> >
> > Regarding memory requirements, our initial platform did not support FlashAttention, impacting our FP16 results. With proper support for FlashAttention now established, we aim to further explore FP16's potential in distilling SiD-LSG, particularly to see if we can match FP32's performance at a lower cost. We have updated Table 4 in the Appendix, showing significant memory reduction and a clear speed up in iterations under FP16. This improvement allows us to opt for larger batch sizes per GPU under FP16, enhancing time efficiency further.
> >
> > We note the current FP32 implementation is based on DDP, which restricts the batch size to one due to memory constraints. A potential improvement could involve adopting another parallel computing framework that allows model splitting across GPUs, potentially leading to larger per-GPU batch sizes and thus improved iteration efficiency. However, implementing these code-level optimization is beyond the scope of this paper.

---

> ### Comment · Reviewer_8grk · 2024-11-19
>
> Thank you the authors for your response. I see better the difference between your long and short CFG mechanism and the one used in SwiftBrush.
>
> 1. So in SwiftBrush, $\kappa_1 = 1$ and $\kappa_2 = \kappa_3 > 1$, while in your method $\kappa_1 = \kappa_2 = \kappa_3 > 1$. I am not sure if the difference has a significant effect. Can you provide the results of your system when using the same cfg mechanism used in SwiftBrush with $\kappa_1 = 1$, $\kappa_2 = \kappa_3 > 1$, $\kappa_4$ is either 1 or equals to $\kappa_2$, which is not provided in your paper?
>
> 2. The further discussion text for the long/short/long and short strategies should be moved to the main text. Also, I still find the definitions of these strategies quite forced.
>
> 3. SwiftBrushv2 has a data-free version with FID 8.77. Its best version, with FID 8.14, only employs a small set of 200k images for regularization, which DMD did. DMD2 starts with a data-free pipeline (before adding the GAN loss), though its data-free version seems to have bad quality.
>
> 4. I see that you can greatly improve the training efficiency with FP16. However, according to Fig. 10 in the Appendix, your FP16 versions have worse FID scores than the FP32 ones. I assume the numbers reported in Table 1 are with FP32. Could you provide the key results with FP16 to see how much performance drops when employing that efficient training?

---

> ### Author Response · Authors · 2024-11-19
>
> Thank you for reviewing our response and posing additional questions, which we are pleased to address.
>
> 1. It is important to highlight that there is no equivalent of $\kappa_3$ in the Diff-Instruct/DMD/SwiftBrush loss frameworks.
>
>
>    At the beginning of our study, we tested various combinations of $\kappa_{1,2,3,4}$ and quickly eliminated those that converged too slowly or diverged. Specifically, the combination $\kappa_1=1$, $\kappa_2=\kappa_3>1$, and $\kappa_4 \in ${$\kappa_2,1$} was discarded early due to poor performance, as depicted in Figure 11 newly added into the Appendix.
>
> 2. We will further clarify the distinctions between long, short, and long-short guidance strategies in our next revision.
>
> 3. We emphasize that SwiftBrushv2 is **not** entirely a data-free method, as it requires more than just access to the teacher model and user-provided text prompts. Unlike SwiftBrush, which is data-free but yields a higher FID, SwiftBrushv2 substantially improves upon its predecessor. However, it achieves this through extensive engineering efforts, including initialization from SD-Turbo—a model trained with adversarial diffusion distillation that requires real data access, thereby disqualifying it as data-free.
>
> -  SwiftBrushv2 employs several enhancements to lower its FID from above 15 to 8.77, and then to 8.14. These modifications could also benefit SiD-LSG, which had already achieved an FID of 8.15 without them:
>      - Leveraging pretrained weights from SD-Turbo to initialize the student network within SwiftBrush’s training framework.
>      - Augmenting its prompts with data not only from the 1.5M Journey DB but also from 2M entries from LAION.
>      - Introducing a clamped CLIP loss to directly boost the CLIP score. Although this undoubtedly increases the CLIP score, it remains uncertain whether it genuinely enhances text-image alignment as significantly as the improved score suggests.
>     - Applying post-training integration of weights from two differently trained models, which involves training the models separately and merging their weights afterward.
>      - Incorporating DMD-like image regularization with a human-feedback dataset of 200K images from LAION-Aesthetic-6.25+ to further decrease the FID from 8.77 to 8.14, making SwiftBrushv2 categorically not data-free.
>
> - We observe that SiD-LSG's FID of 8.15 not only outperforms DMD but also DMD2, which has an FID of 8.34. DMD2 introduces several techniques to enhance DMD, these could potentially benefit SiD-LSG as well.
>
> - In summary, SwiftBrushv2 and DMD2, which we regard as concurrent works, explore enhancements to their base models derived from Diff-Instruct/variational score distillation by incorporating various additional techniques. In contrast, SiD-LSG concentrates on elevating the base data-free SiD model, derived from a mode-based Fisher divergence, to a strong contender in the field of text-to-image generation, without relying on these extra efforts. Consequently, SiD-LSG and the approaches used in SwiftBrushv2 and DMD2 are naturally complementary to each other.
>
> 4. Regarding precision and computational efficiency, although we have achieved state-of-the-art results under FP32 with acceptable computing demand, our experiments with FP16 showed faster convergence but typically resulted in FID scores more than 2 points worse than those achieved with FP32. Our current hypothesis is that adjusting the loss scaling for both the generator and the fake score network might help prevent numerical under/overflow, commonly encountered in FP16 environments. Presently, we have not applied loss scaling. Alternatively, employing PyTorch autocast and GradScaler for mixed precision could potentially enhance performance. We consider these adjustments less critical at this stage but plan to explore them in future research. Additionally, we observe that using FP16 for inference, as is standard in SiD-LSG when reporting FID and CLIP scores, does not adversely affect performance.

---

> > ### Comment · Reviewer_8grk · 2024-11-19
> >
> > Thank you for your response. While I still find the definitions of the long/short strategies a bit forced, and there is a performance gap between the efficient model training and the expensive FP32 one, I am okay with accepting the paper due to its good results and extensive analyses. I have increased my score.

---

> > > ### Author Response · Authors · 2024-11-19
> > >
> > > Thank you! We are pleased to have obtained your support.
> > >
> > > - The discussions have been instrumental in helping us understand how to better articulate our ideas. Specifically, to better explain the final recommended LSG setting, we will clearly describe the enhancement of the teacher's guidance as "long," while referring to the same CFG scale—but with effectively less enhancement—of the fake score's guidance during evaluation as "short."
> > >
> > > - From my experience, the performance gap is likely addressable through a grid search of loss scaling or by maintaining only certain critical components in FP32. We are committed to exploring these possibilities in our future research. If FP16 can match the performance of FP32 for SiD-LSG, it would become an even more compelling solution, especially since it is at least six times faster with the aid of FlashAttention.

---

### Official Review · Reviewer_i638 · 2024-11-04

**Soundness:** 3
**Presentation:** 3
**Contribution:** 2
**Rating:** 6
**Confidence:** 3

**Summary:**

This paper focuses on a novel approach to accelerating text-to-image diffusion models through Score Identity Distillation with Long-Short Guidance (SiD-LSG), specifically targeting data-free one-step generation. By combining Score Identity Distillation with Classifier-Free Guidance (CFG), the authors efficiently distill Stable Diffusion models into one-step generators without needing real data. The proposed method incorporates both long and short CFG strategies, which balance FID and CLIP scores, enabling efficient and high-quality text-to-image generation. Experimental results demonstrate the effectiveness of the proposed method on the COCO dataset, achieving a record-low data-free FID of 8.15.

**Strengths:**

The paper introduces two distinct strategies for CFG in SiD: the long CFG and short CFG strategies, each with unique effects on model performance.
1. In the long CFG, it is applied to the pretrained score network (the teacher model) with a higher guidance scale (typically $\tau$>1), which encourages the pretrained model to generate images more aligned with the textual prompt, which in turn compels the student generator to follow this alignment. The primary effect is a strong improvement in text adherence and quality, with the capability to achieve a balanced FID and CLIP score.
2. In the short CFG, it is applied to the fake score network (the student model) with a reduced guidance scale (typically $\tau$ values between 0 and 1). By reducing CFG, this approach lessens the fake score network’s alignment with the text prompt, which incentivizes the student generator to improve text alignment and quality. This strategy tends to result in a trade-off where lower CLIP scores (text alignment) can occur, but FID scores (distribution match) remain competitive.

**Weaknesses:**

1. The main contribution of this paper is introducing a variant of CFG into SiD, i.e., long and short CFGs, which aim to balance semantic alignment and the quality of generated images. However, there seems to be a contradiction, as no single $\tau$ value optimally balances both CLIP and FID scores (see results in Tables 1 and 2). Are there intrinsic reasons for this phenomenon?

2. A more intuitive ablation study would better showcase the effectiveness of the proposed long and short CFG strategies. Specifically, in Tables 1 and 2, rather than using SiD-LSG alone, could you provide the results (both CLIP and FID) for SiD with the same $\tau$ value across different configurations: SiD (short CFG only), SiD (long CFG only), SiD (CFG only), and SiD (original)?

**Questions:**

Please refer to the Weaknesses.

---

> ### Author Response · Authors · 2024-11-17
> **Conflict between High CLIP and Low FID and Area under the CLIP-FID curve**
>
> We appreciate the reviewer's insightful summary and detailed discussion on both the long and short CFG strategies. We address the raised points as follows:
>
> **Point 1:** A prevailing challenge in current text-to-image diffusion models and their distilled variants lies in optimizing distribution matching—characterized by low FID and robust generation diversity—and aligning generated images with provided text prompts, which is reflected in high CLIP scores. In existing literature, it is not uncommon to emphasize the FID for models trained or inferred under a low CFG scale, while using a model under a high CFG to report CLIP and visualize generations. This practice, which we explicitly avoided, often obscures the true performance trade-offs.
>
> In our research, we consistently report both FID and CLIP for every model we distill, directly addressing this contradiction. We introduce Long-Short Guidance (LSG) as a method to mitigate, although not entirely resolve, this issue. Our results demonstrate that LSG achieves a more balanced performance between low FID and high CLIP compared to both long and short CFG strategies.
>
> The underlying causes of this contradiction likely differ among model types:
> - **Teacher Diffusion Models (e.g., SD1.5 and SD2.1-base):** The high-guidance scale during inference forces these models to concentrate on high-density regions of the data distribution conditioned on text prompts, achieving lower FIDs but at the expense of reduced diversity and poorer distribution matching in sparser regions.
> - **Data-Free Distillation Methods:** Typically using long CFG strategies, these models emulate the behavior of the CFG-enhanced teacher, leading to similar challenges.
> - **Adversarial Boosted Distillation Methods (e.g., ADD by Sauer et al., 2023b):** These methods typically perform clearly worse without their adversarial components. They are prone to adversarial loss inducing mode collapse or seeking behaviors, which can significantly compromise diversity and negatively impact FID.
>
> **Point 2:** We thank the reviewer for their valuable suggestions. To more effectively demonstrate the effectiveness of our proposed LSG, we have incorporated Table 3 and Figure 9 in the Appendix. Drawing an analogy between CLIP vs. FID and the True Positive Rate vs. False Positive Rate in a Receiver Operating Characteristic (ROC) curve, Figure 9 illustrates that LSG achieves the largest Area Under the CLIP-FID Curve. This underscores its superiority over both the Long and Short CFG strategies.
>
> LSG applies CFGs to the evaluation of the teacher and to both the training and evaluation of the fake score network, guiding the learning of the student generator. As CFGs are not utilized in reverse sampling but only in guiding the learning of the fake score and the single-step generators, they provide opportunities to better preserve the matching of the data distribution.

---

> ### Comment · Reviewer_i638 · 2024-11-27
> **Reply to authors**
>
> Thank you for the author’s response, which addressed many of my concerns. I’ve decided to keep the score.

---

### Meta-Review · Area_Chair_85D4 · 2024-12-23

**Metareview:**

This paper focuses on accelerating diffusion-based generative models. It introduces a novel method, Guided Score Identity Distillation with Long-Short Guidance (SiD-LSG), designed for data-free one-step text-to-image generation. The approach builds upon the existing Score Identity Distillation (SiD) framework by integrating Long and Short Classifier-Free Guidance (LSG) to enhance its effectiveness.

A key strength of this work lies in its data-free nature, making it much more generalizable especially when access to real training data is limited. The experimental results also demonstrate state-of-the-art performance. The method's scalability and adaptability to various diffusion models further enhance its significance.

Given these strengths and the consistent positive feedback from the reviewers, I recommend acceptance of the paper.

**Additional Comments On Reviewer Discussion:**

Initially, one major concern was the novelty of the paper, particularly in comparison to SwiftBrush and DMD. During the rebuttal, the authors addressed this issue by highlighting that, unlike SwiftBrush and DMD, which rely on real data, SiD-LSG operates entirely in a data-free setting. This clarification successfully resolved the concern regarding the paper's novelty.

---

### Decision · Program_Chairs · 2025-01-22

Accept (Poster)